# Rules and mechanisms for efficient two-stage learning in neural circuits

**Tiberiu Teşileanu[1,2], Bence Ölveczky[3], Vijay Balasubramanian[1,2,4]***

[1]Initiative for the Theoretical Sciences, CUNY Graduate Center, New York, United States; [2]David Rittenhouse Laboratories, University of Pennsylvania, Philadelphia, United States; [3]Department of Organismic and Evolutionary Biology and Center for Brain Science, Harvard University, Cambridge, United States; [4]Theoretische Natuurkunde, Vrije Universiteit Brussel & International Solvay Institutes, Brussels, Belgium

**Abstract** Trial-and-error learning requires evaluating variable actions and reinforcing successful variants. In songbirds, vocal exploration is induced by LMAN, the output of a basal ganglia-related circuit that also contributes a corrective bias to the vocal output. This bias is gradually consolidated in RA, a motor cortex analogue downstream of LMAN. We develop a new model of such two-stage learning. Using stochastic gradient descent, we derive how the activity in 'tutor' circuits (*e.g.*, LMAN) should match plasticity mechanisms in 'student' circuits (*e.g.*, RA) to achieve efficient learning. We further describe a reinforcement learning framework through which the tutor can build its teaching signal. We show that mismatches between the tutor signal and the plasticity mechanism can impair learning. Applied to birdsong, our results predict the temporal structure of the corrective bias from LMAN given a plasticity rule in RA. Our framework can be applied predictively to other paired brain areas showing two-stage learning.

*For correspondence: vijay@physics.upenn.edu

**Competing interests:** The authors declare that no competing interests exist.

## Introduction

Two-stage learning has been described in a variety of different contexts and neural circuits. During hippocampal memory consolidation, recent memories, that are dependent on the hippocampus, are transferred to the neocortex for long-term storage (*Frankland and Bontempi, 2005*). Similarly, the rat motor cortex provides essential input to sub-cortical circuits during skill learning, but then becomes dispensable for executing certain skills (*Kawai et al., 2015*). A paradigmatic example of two-stage learning occurs in songbirds learning their courtship songs (*Andalman and Fee, 2009*; *Turner and Desmurget, 2010*; *Warren et al., 2011*). Zebra finches, commonly used in birdsong research, learn their song from their fathers as juveniles, and keep the same song for life (*Immelmann, 1969*).

The birdsong circuit has been extensively studied; see *Figure 1A* for an outline. Area HVC is a timebase circuit, with projection neurons that fire sparse spike bursts in precise synchrony with the song (*Hahnloser et al., 2002*; *Lynch et al., 2016*; *Picardo et al., 2016*). A population of neurons from HVC projects to the robust nucleus of the arcopallium (RA), a pre-motor area, which then projects to motor neurons controlling respiratory and syringeal muscles (*Leonardo and Fee, 2005*; *Simpson and Vicario, 1990*; *Yu and Margoliash, 1996*). A second input to RA comes from the lateral magnocellular nucleus of the anterior nidopallium (LMAN). Unlike HVC and RA activity patterns, LMAN spiking is highly variable across different renditions of the song (*Kao et al., 2008*; *Ölveczky et al., 2005*). LMAN is the output of the anterior forebrain pathway, a circuit involving the song-specialized basal ganglia (*Perkel, 2004*).

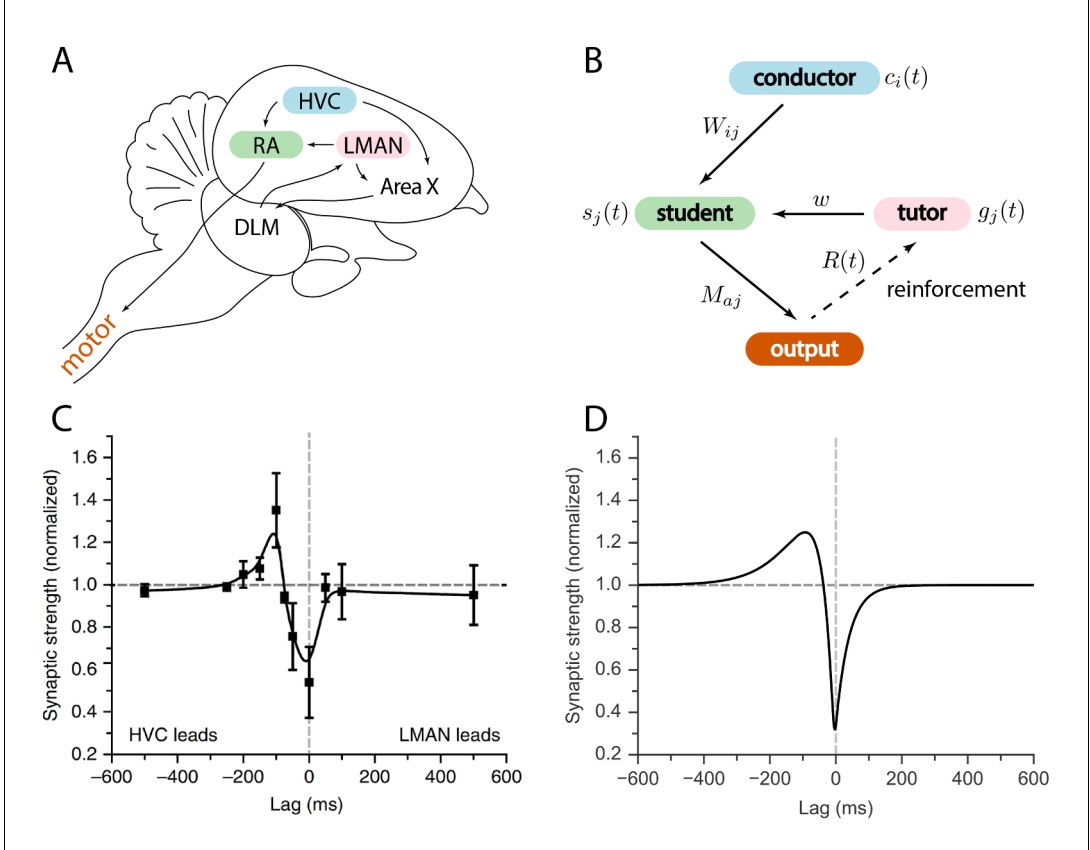

**Figure 1.** Relation between the song system in zebra finches and our model. (**A**) Diagram of the major brain regions involved in birdsong. (**B**) Conceptual model inspired by the birdsong system. The line from output to tutor is dashed because the reinforcement signal can reach the tutor either directly or, as in songbirds, indirectly. (**C**) Plasticity rule measured in bird RA (measurement done in slice). When an HVC burst leads an LMAN burst by about $100\,\mathrm{ms}$, the HVC–RA synapse is strengthened, while coincident firing leads to suppression. Figure adapted from *Mehaffey and Doupe (2015)*. (**D**) Plasticity rule in our model that mimics the *Mehaffey and Doupe (2015)* rule.

Because of the variability in its activity patterns, it was thought that LMAN's role was simply to inject variability into the song (*Ölveczky et al., 2005*). The resulting vocal experimentation would enable reinforcement-based learning. For this reason, prior models tended to treat LMAN as a pure Poisson noise generator, and assume that a reward signal is received directly in RA (*Fiete et al., 2007*). More recent evidence, however, suggests that the reward signal reaches Area X, the song-specialized basal ganglia, rather than RA (*Gadagkar et al., 2016*; *Hoffmann et al., 2016*; *Kubikova et al., 2010*). Taken together with the fact that LMAN firing patterns are not uniformly random, but rather contain a corrective bias guiding plasticity in RA (*Andalman and Fee, 2009*; *Warren et al., 2011*), this suggests that we should rethink our models of song acquisition.

Here we build a general model of two-stage learning where one neural circuit 'tutors' another. We develop a formalism for determining how the teaching signal should be adapted to a specific plasticity rule, to best instruct a student circuit to improve its performance at each learning step. We develop analytical results in a rate-based model, and show through simulations that the general findings carry over to realistic spiking neurons. Applied to the vocal control circuit of songbirds, our model reproduces the observed changes in the spiking statistics of RA neurons as juvenile birds learn their song. Our framework also predicts how the LMAN signal should be adapted to properties of RA synapses. This prediction can be tested in future experiments.

Our approach separates the mechanistic question of *how* learning is implemented from what the resulting learning rules are. We nevertheless demonstrate that a simple reinforcement learning algorithm suffices to implement the learning rule we propose. Our framework makes general predictions

for how instructive signals are matched to plasticity rules whenever information is transferred between different brain regions.

## Results

### Model

We considered a model for information transfer that is composed of three sub-circuits: a conductor, a student, and a tutor (see *Figure 1B*). The conductor provides input to the student in the form of temporally precise patterns. The goal of learning is for the student to convert this input to a prede-fined output pattern. The tutor provides a signal that guides plasticity at the conductor–student synapses. For simplicity, we assumed that the conductor always presents the input patterns in the same order, and without repetitions. This allowed us to use the time $t$ to label input patterns, making it easier to analyze the on-line learning rules that we studied. This model of learning is based on the logic implemented by the vocal circuits of the songbird (*Figure 1A*). Relating this to the songbird, the conductor is HVC, the student is RA, and the tutor is LMAN. The song can be viewed as a mapping between clock-like HVC activity patterns and muscle-related RA outputs. The goal of learning is to find a mapping that reproduces the tutor song.

Birdsong provides interesting insights into the role of variability in tutor signals. If we focus solely on information transfer, the tutor output need not be variable; it can deterministically provide the best instructive signal to guide the student. This, however, would require the tutor to have a detailed model of the student. More realistically, the tutor might only have access to a scalar representation of how successful the student rendition of the desired output is, perhaps in the form of a reward signal. A tutor in this case has to solve the so-called 'credit assignment problem'—it needs to identify which student neurons are responsible for the reward. A standard way to achieve this is to inject variability in the student output and reinforce the firing of neurons that precede reward (see for example (*Fiete et al., 2007*) in the birdsong context). Thus, in our model, the tutor has a dual role of providing both an instructive signal and variability, as in birdsong.

We described the output of our model using a vector $y_a(t)$ where $a$ indexed the various output channels (*Figure 2A*). In the context of motor control $a$ might index the muscle to be controlled, or, more abstractly, different features of the motor output, such as pitch and amplitude in the case of birdsong. The output $y_a(t)$ was a function of the activity of the student neurons $s_j(t)$. The student neurons were in turn driven by the activity of the conductor neurons $c_i(t)$. The student also received tutor signals to guide plasticity; in the songbird, the guiding signals for each RA neuron come from several LMAN neurons (*Canady et al., 1988*; *Garst-Orozco et al., 2014*; *Herrmann and Arnold, 1991*). In our model, we summarized the net input from the tutor to the $j$th student neuron as a single function $g_j(t)$.

We started with a rate-based implementation of the model (*Figure 2A*) that was analytically tractable but averaged over tutor variability. We further took the neurons to be in a linear operating regime (*Figure 2A*) away from the threshold and saturation present in real neurons. We then relaxed these conditions and tested our results in spiking networks with initial parameters selected to imitate measured firing patterns in juvenile birds prior to song learning. The student circuit in both the rate-based and spiking models included a global inhibitory signal that helped to suppress excess activity driven by ongoing conductor and tutor input. Such recurrent inhibition is present in area RA of the bird (*Spiro et al., 1999*). In the spiking model we implemented the suppression as an activity-dependent inhibition, while for the analytic calculations we used a constant negative bias for the student neurons.

### Learning in a rate-based model

Learning in our model was enabled by plasticity at the conductor–student synapses that was modulated by signals from tutor neurons (*Figure 2B*). Many different forms of such hetero-synaptic plasticity have been observed. For example, in rate-based synaptic plasticity high tutor firing rates lead to synaptic potentiation and low tutor firing rates lead to depression (*Chistiakova and Volgushev, 2009*; *Chistiakova et al., 2014*). In timing-dependent rules, such as the one recently measured by *Mehaffey and Doupe (2015)* in slices of zebra finch RA (see *Figure 1C*), the relative arrival times of spike bursts from different input pathways set the sign of synaptic change. To model learning that

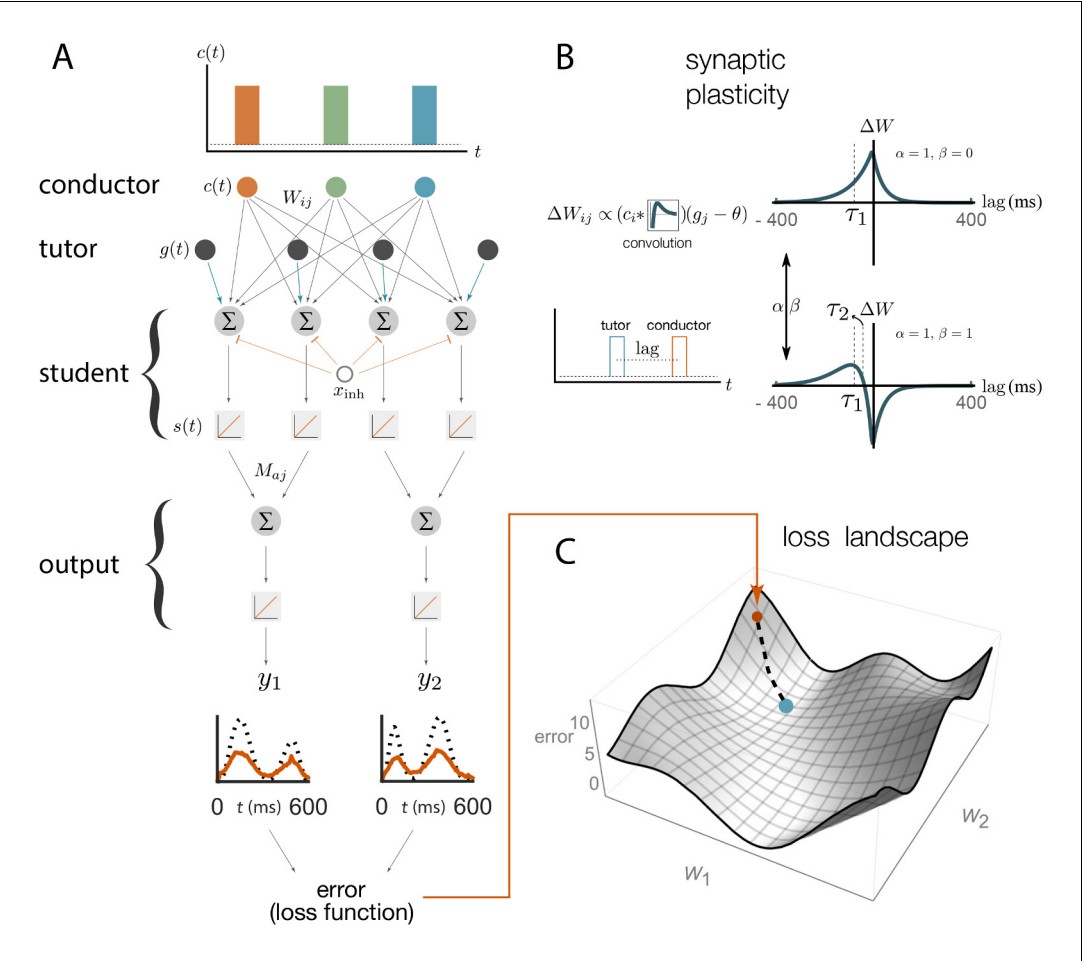

**Figure 2.** Schematic representation of our rate-based model. (A) Conductor neurons fire precisely-timed bursts, similar to HVC neurons in songbirds. Conductor and tutor activities, $c(t)$ and $g(t)$, provide excitation to student neurons, which integrate these inputs and respond linearly, with activity $s(t)$. Student neurons also receive a constant inhibitory input, $x_{\text{inh}}$. The output neurons linearly combine the activities from groups of student neurons using weights $M_{aj}$. The linearity assumptions were made for mathematical convenience but are not essential for our qualitative results (see Appendix). (B). The conductor–student synaptic weights $W_{ij}$ are updated based on a plasticity rule that depends on two parameters, $\alpha$ and $\beta$, and two timescales, $\tau_1$ and $\tau_2$ (see *Equation (1)* and Materials and methods). The tutor signal enters this rule as a deviation from a constant threshold $\theta$. The figure shows how synaptic weights change ($\Delta W$) for a student neuron that receives a tutor burst and a conductor burst separated by a short lag. Two different choices of plasticity parameters are illustrated in the case when the threshold $\theta = 0$. (C) The amount of mismatch between the system's output and the target output is quantified using a loss (error) function. The figure sketches the loss landscape obtained by varying the synaptic weights $W_{ij}$ and calculating the loss function in each case (only two of the weight axes are shown). The blue dot shows the lowest value of the loss function, corresponding to the best match between the motor output and the target, while the orange dot shows the starting point. The dashed line shows how learning would proceed in a gradient descent approach, where the weights change in the direction of steepest descent in the loss landscape.

lies between these rate and timing-based extremes, we introduced a class of plasticity rules governed by two parameters $\alpha$ and $\beta$ (see also Materials and methods and *Figure 2B*):

$$\frac{dW_{ij}}{dt} = \eta \tilde{c}_i(t)\left(g_j(t) - \theta\right),$$

$$\tilde{c}_i(t) = \int_0^t dt' c_i(t')\left[\frac{\alpha}{\tau_1}e^{-(t-t')/\tau_1} - \frac{\beta}{\tau_2}e^{-(t-t')/\tau_2}\right], \tag{1}$$

where $W_{ij}$ is the weight of the synapse from the $i$th conductor to the $j$th student neuron, $\eta$ is a learning rate, $\theta$ is a threshold on the firing rate of tutor neurons, and $\tau_1$ and $\tau_2$ are timescales associated with the plasticity. This is similar to an STDP rule, except that the dependence on postsynaptic

activity was replaced by dependence on the input from the tutor. Thus plasticity acts heterosynaptically, with activation of the tutor–student synapse controlling the change in the conductor–student synaptic weight. The timescales $\tau_1$ and $\tau_2$, as well as the coefficients $\alpha$ and $\beta$, can be thought of as effective parameters describing the plasticity observed in student neurons. As such, they do not necessarily have a simple correspondence in terms of the biochemistry of the plasticity mechanism, and the framework we describe here is not specifically tied to such an interpretation.

If we set $\alpha$ or $\beta$ to zero in our rule, *Equation (1)*, the sign of the synaptic change is determined solely by the firing rate of the tutor $g_j(t)$ as compared to a threshold, reproducing the rate rules observed in experiments. When $\alpha/\beta \approx 1$, if the conductor leads the tutor, potentiation occurs, while coincident signals lead to depression (*Figure 2B*), which mimics the empirical findings from *Mehaffey and Doupe (2015)*. For general $\alpha$ and $\beta$, the sign of plasticity is controlled by both the firing rate of the tutor relative to the baseline, and by the relative timing of tutor and conductor. The overall scale of the parameters $\alpha$ and $\beta$ can be absorbed into the learning rate $\eta$ and so we set $\alpha - \beta = 1$ in all our simulations without loss of generality (see Materials and methods). Note that if $\alpha$ and $\beta$ are both large, it can be that $\alpha - \beta = 1$ and $\alpha/\beta \approx 1$ also, as needed to realize the *Mehaffey and Doupe (2015)* curve.

We can ask how the conductor–student weights $W_{ij}$ (*Figure 2A*) should change in order to best improve the output $y_a(t)$. We first need a loss function $L$ that quantifies the distance between the current output $y_a(t)$ and the target $\bar{y}_a(t)$ (*Figure 2C*). We used a quadratic loss function, but other choices can also be incorporated into our framework (see Appendix). Learning should change the synaptic weights so that the loss function is minimized, leading to a good rendition of the targeted output. This can be achieved by changing the synaptic weights in the direction of steepest descent of the loss function (*Figure 2C*).

We used the synaptic plasticity rule from *Equation (1)* to calculate the overall change of the weights, $\Delta W_{ij}$, over the course of the motor program. This is a function of the time course of the tutor signal, $g_j(t)$. Not every choice for the tutor signal leads to motor output changes that best improve the match to the target. Imposing the condition that these changes follow the gradient descent procedure described above, we derived the tutor signal that was best matched to the student plasticity rule (detailed derivation in Materials and methods). The result is that the best tutor for driving gradient descent learning must keep track of the motor error

$$\epsilon_j(t) = \sum_a M_{aj}(y_a(t) - \bar{y}_a(t)) \tag{2}$$

integrated over the recent past

$$g_j(t) = \theta - \frac{\zeta}{\alpha - \beta} \frac{1}{\tau_{\text{tutor}}} \int_0^t \epsilon_j(t') e^{-(t-t')/\tau_{\text{tutor}}} \, dt', \tag{3}$$

where $M_{aj}$ are the weights describing the linear relationship between student activities and motor outputs (*Figure 2A*) and $\zeta$ is a learning rate. Moreover, for effective learning, the parameter $\tau_{\text{tutor}}$ appearing in *Equation (3)*, which quantifies the timescale on which error information is integrated into the tutor signal, should be related to the synaptic plasticity parameters according to

$$\begin{aligned} \tau_{\text{tutor}} &= \tau_{\text{tutor}}^*, \quad \text{where} \\ \tau_{\text{tutor}}^* &\equiv \frac{\alpha \tau_1 - \beta \tau_2}{\alpha - \beta} \end{aligned} \tag{4}$$

is the optimal timescale for the error integration.

In short, motor learning with a heterosynaptic plasticity rule requires convolving the motor error with a kernel whose timescale is related to the structure of the plasticity rule, but is otherwise independent of the motor program. As explained in more detail in Materials and methods, this result is derived in an approximation that assumes that the tutor signal does not vary significantly over timescales of the order of the student timescales $\tau_1$ and $\tau_2$. Given *Equation (4)*, this implies that we are assuming $\tau_{\text{tutor}} \gg \tau_{1,2}$. This is a reasonable approximation because variations in the tutor signal that are much faster than the student timescales $\tau_{1,2}$ have little effect on learning since the plasticity rule (1) blurs conductor inputs over these timescales.

## Matched *vs.* unmatched learning

Our rate-based model predicts that when the timescale on which error information is integrated into the tutor signal ($\tau_{\mathrm{tutor}}$) is matched to the student plasticity rule as described above, learning will proceed efficiently. A mismatched tutor should slow or disrupt convergence to the desired output. To test this, we numerically simulated the birdsong circuit using the linear model from *Figure 2A* with a motor output $y_a$ filtered to more realistically reflect muscle response times (see Materials and methods). We selected plasticity rules as described in *Equation (1)* and *Figure 2B* and picked a target output pattern to learn. The target was chosen to resemble recordings of air-sac pressure from singing zebra finches in terms of smoothness and characteristic timescales (*Veit et al., 2011*), but was otherwise arbitrary. In our simulations, the output typically involved two different channels, each with its own target, but for brevity, in figures we typically showed the output from only one of these.

For our analytical calculations, we made a series of assumptions and approximations meant to enhance tractability, such as linearity of the model and a focus on the regime $\tau_{\mathrm{tutor}} \gg \tau_{1,2}$. These constraints can be lifted in our simulations, and indeed below we test our numerical model in regimes that go beyond the approximations made in our derivation. In many cases, we found that the basic findings regarding tutor–student matching from our analytical model remain true even when some of the assumptions we used to derive it no longer hold.

We tested tutors that were matched or mismatched to the plasticity rule to see how effectively they instructed the student. *Figure 3A* and online *Video 1* show convergence with a matched tutor when the sign of plasticity is determined by the tutor's firing rate. We see that the student output rapidly converged to the target. *Figure 3B* and online *Video 2* show convergence with a matched tutor when the sign of plasticity is largely determined by the relative timing of the tutor signal and the student output. We see again that the student converged steadily to the desired output, but at a somewhat slower rate than in *Figure 3A*.

To test the effects of mismatch between tutor and student, we used tutors with timescales that did not match *Equation (4)*. All student plasticity rules had the same effective time constants $\tau_1$ and $\tau_2$, but different parameters $\alpha$ and $\beta$ (see *Equation 1*), subject to the constraint $\alpha - \beta = 1$ described in the previous section. Different tutors had different memory time scales $\tau_{\mathrm{tutor}}$ (*Equation 3*). *Figure 3C and D* demonstrate that learning was more rapid for well-matched tutor-student pairs (the diagonal neighborhood, where $\tau_{\mathrm{tutor}} \approx \tau_{\mathrm{tutor}}^*$). When the tutor error integration timescale was shorter than the matched value in *Equation (4)*, $\tau_{\mathrm{tutor}} < \tau_{\mathrm{tutor}}^*$, learning was often completely disrupted (many pairs below the diagonal in *Figure 3C and D*). When the tutor error integration timescale was longer than the matched value in *Equation (4)*, $\tau_{\mathrm{tutor}} > \tau_{\mathrm{tutor}}^*$ learning was slowed down. *Figure 3C* also shows that a certain amount of mismatch between the tutor error integration timescale $\tau_{\mathrm{tutor}}$ and the matched timescale $\tau_{\mathrm{tutor}}^*$ implied by the student plasticity rule is tolerated by the system. Interestingly, the diagonal band over which learning is effective in *Figure 3C* is roughly of constant width—note that the scale on both axes is logarithmic, so that this means that the tutor error integration timescale $\tau_{\mathrm{tutor}}$ has to be within a constant factor of the optimal timescale $\tau_{\mathrm{tutor}}^*$ for good learning. We also see that the breakdown in learning is more abrupt when $\tau_{\mathrm{tutor}} < \tau_{\mathrm{tutor}}^*$ than in the opposite regime.

An interesting feature of the results from *Figure 3C and D* is that the difference in performance between matched and mismatched pairs becomes less pronounced for timescales shorter than about $100 \, \mathrm{ms}$. This is due to the fact that the plasticity rule (*Equation 1*) implicitly smooths over timescales of the order of $\tau_{1,2}$, which in our simulations were equal to $\tau_1 = 80 \, \mathrm{ms}$, $\tau_2 = 40 \, \mathrm{ms}$. Thus, variations of the tutor signal on shorter timescales have little effect on learning. Using different values for the effective timescales $\tau_{1,2}$ describing the plasticity rule can increase or decrease the range of parameters over which learning is robust against tutor–student mismatches (see Appendix).

## Robust learning with nonlinearities

In the model above, firing rates for the tutor were allowed to grow as large as necessary to implement the most efficient learning. However, the firing rates of realistic neurons typically saturate at some fixed bound. To test the effects of this nonlinearity in the tutor, we passed the ideal tutor activity (*Equation 3*) through a sigmoidal nonlinearity,

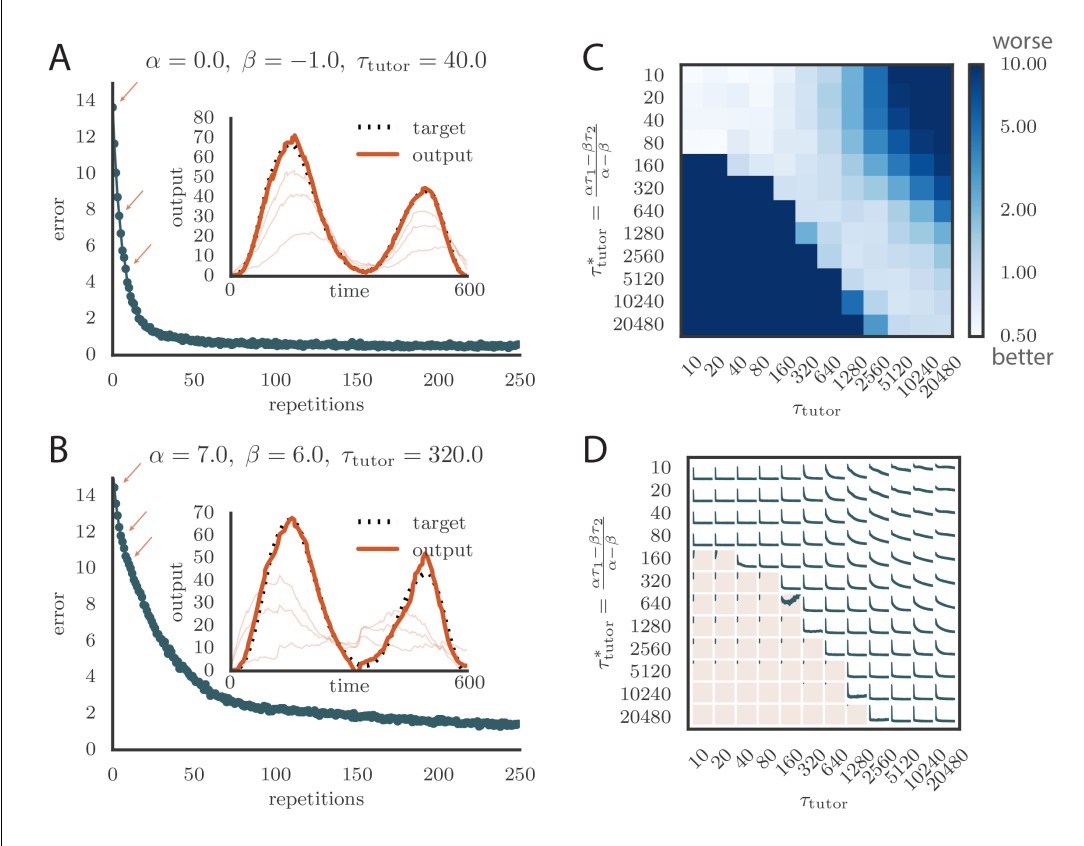

**Figure 3.** Learning with matched or mismatched tutors in rate-based simulations. (**A**) Error trace showing how the average motor error evolved with the number of repetitions of the motor program for a rate-based ($\alpha = 0$) plasticity rule paired with a matching tutor. (See online **Video 1**). (**B**) The error trace and final motor output shown for a timing-based learning rule matched by a tutor with a long integration timescale. (See online **Video 2**.) In both A and B the inset shows the final motor output for one of the two output channels (thick orange line) compared to the target output for that channel (dotted black line). The output on the first rendition and at two other stages of learning indicated by orange arrows on the error trace are also shown as thin orange lines. (**C**) Effects of mismatch between student and tutor on reproduction accuracy. The heatmap shows the final reproduction error of the motor output after 1000 learning cycles in a rate-based simulation where a student with parameters $\alpha$, $\beta$, $\tau_1$, and $\tau_2$ was paired with a tutor with memory timescale $\tau_{\text{tutor}}$. On the $y$ axis, $\tau_1$ and $\tau_2$ were kept fixed at $80\,\text{ms}$ and $40\,\text{ms}$, respectively, while $\alpha$ and $\beta$ were varied (subject to the constraint $\alpha - \beta = 1$; see text). Different choices of $\alpha$ and $\beta$ lead to different optimal timescales $\tau^*_{\text{tutor}}$ according to **Equation (4)**. The diagonal elements correspond to matched tutor and student, $\tau_{\text{tutor}} = \tau^*_{\text{tutor}}$. Note that the color scale is logarithmic. (**D**) Error evolution curves as a function of the mismatch between student and tutor. Each plot shows how the error in the motor program changed during 1000 learning cycles for the same conditions as those shown in the heatmap. The region shaded in light pink shows simulations where the mismatch between student and tutor led to a deteriorating instead of improving performance during learning.

$$\tilde{g}_j(t) = \theta - \rho \tanh \frac{\zeta}{\alpha - \beta} \frac{1}{\tau_{\text{tutor}}} \int_0^t \epsilon_j(t') e^{-(t-t')/\tau_{\text{tutor}}} \, dt' \, . \tag{5}$$

where $2\rho$ is the range of firing rates. We typically chose $\theta = \rho = 80\,\text{Hz}$ to constrain the rates to the range 0–160 Hz (**Ölveczky et al., 2005**; **Garst-Orozco et al., 2014**). Learning slowed down with this change (**Figure 4A** and online **Video 3**) as a result of the tutor firing rates saturating when the mismatch between the motor output and the target output was large. However, the accuracy of the final rendition was not affected by saturation in the tutor (**Figure 4A**, inset). An interesting effect occurred when the firing rate constraint was imposed on a matched tutor with a long memory timescale. When this happened and the motor error was large, the tutor signal saturated and stopped growing in relation to the motor error before the end of the motor program. In the extreme case of very long integration timescales, learning became sequential: early features in the output were

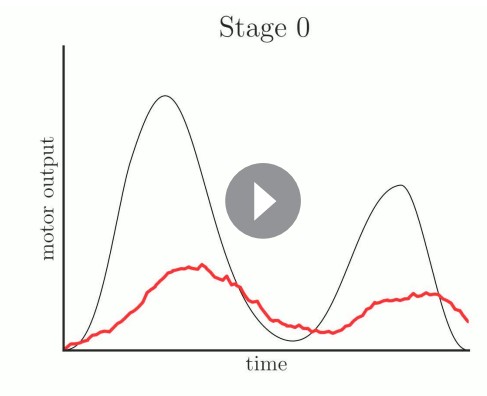

**Video 1.** Evolution of motor output during learning in a rate-based simulation using a rate-based ($\alpha = 0$) plasticity rule paired with a matching tutor. This video relates to *Figure 3A*.

learned first, before later features were addressed, as in *Figure 4B* and online *Video 4*. This is reminiscent of the learning rule described in (*Memmesheimer et al., 2014*).

Nonlinearities can similarly affect the activities of student neurons. Our model can be readily extended to describe efficient learning even in this case. The key result is that for efficient learning to occur, the synaptic plasticity rule should depend not just on the tutor and conductor, but also on the activity of the postsynaptic student neurons (details in Appendix). Such dependence on postsynaptic activity is commonly seen in experiments (*Chistiakova and Volgushev, 2009*; *Chistiakova et al., 2014*).

The relation between student neuron activations $s_j(t)$ and motor outputs $y_a(t)$ (*Figure 2A*) is in general also nonlinear. Compared to the linear assumption that we used, the effect of a monotonic nonlinearity, $y_a = N_a(\sum_j M_{aj}s_j)$, with $N_a$ an increasing function, is similar to modifying the loss function $L$, and does not significantly change our results (see Appendix). We also checked that imposing a rectification constraint that conductor–student weights $W_{ij}$ must be positive does not modify our results either (see Appendix). This shows that our model continues to work with biologically realistic synapses that cannot change sign from excitatory to inhibitory during learning.

## Spiking neurons and birdsong

To apply our model to vocal learning in birds, we extended our analysis to networks of spiking neurons. Juvenile songbirds produce a 'babble' that converges through learning to an adult song strongly resembling the tutor song. This is reflected in the song-aligned spiking patterns in premotor area RA, which become more stereotyped and cluster in shorter, better-defined bursts as the bird matures (*Figure 5A*). We tested whether our model could reproduce key statistics of spiking in RA over the course of song learning. In this context, our theory of efficient learning, derived in a rate-based scenario, predicts a specific relation between the teaching signal embedded in LMAN firing patterns, and the plasticity rule implemented in RA. We tested whether these predictions continued to hold in the spiking context.

Following the experiments of *Hahnloser et al. (2002)*, we modeled each neuron in HVC (the conductor) as firing one short, precisely timed burst of 5–6 spikes at a single moment in the motor program. Thus the population of HVC neurons produced a precise timebase for the song. LMAN (tutor) neurons are known to have highly variable firing patterns that facilitate experimentation, but also contain a corrective bias (*Andalman and Fee, 2009*). Thus we modeled LMAN as producing inhomogeneous Poisson spike trains with a time-dependent firing rate given by *Equation (5)* in our model. Although biologically there are several LMAN neurons projecting to each RA neuron, we again simplified by 'summing' the LMAN inputs into a single, effective tutor neuron, similarly to the approach in (*Fiete et al., 2007*). The LMAN-RA synapses were modeled in a current-based

**Video 2.** Evolution of motor output during learning in a rate-based simulation using a timing-based ($\alpha \approx \beta$) plasticity rule paired with a matching tutor. This video relates to *Figure 3B*.

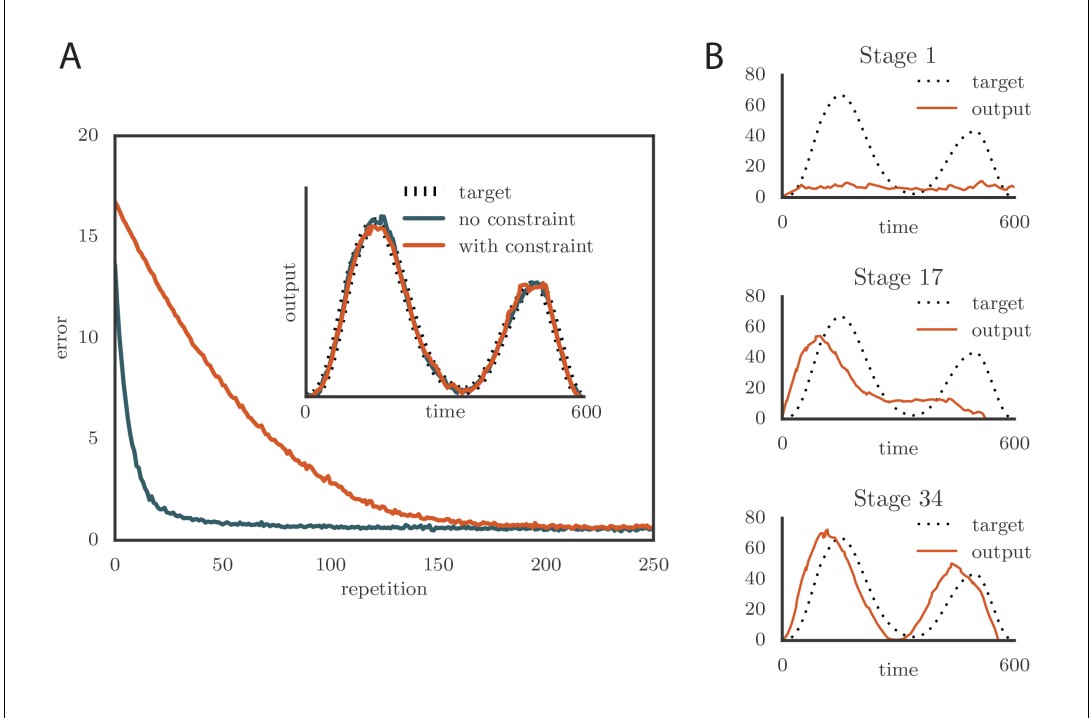

**Figure 4.** Effects of adding a constraint on the tutor firing rate to the simulations. (**A**) Learning was slowed down by the firing rate constraint, but the accuracy of the final rendition stayed the same (inset, shown here for one of two simulated output channels). Here $\alpha = 0$, $\beta = -1$, and $\tau_{\text{tutor}} = \tau_{\text{tutor}}^* = 40\,\text{ms}$. (See online **Video 3**.) (**B**) Sequential learning occurred when the firing rate constraint was imposed on a matched tutor with a long memory scale. The plots show the evolution of the motor output for one of the two channels that were used in the simulation. Here $\alpha = 24$, $\beta = 23$, and $\tau_{\text{tutor}} = \tau_{\text{tutor}}^* = 1000\,\text{ms}$. (See online **Video 4**.).

approach as a mixture of AMPA and NMDA receptors, following the songbird data (*Garst-Orozco et al., 2014*; *Stark and Perkel, 1999*). The initial weights for all synapses were tuned to produce RA firing patterns resembling juvenile birds (*Ölveczky et al., 2011*), subject to constraints from direct measurements in slice recordings (*Garst-Orozco et al., 2014*) (see Materials and methods for details, and *Figure 5B* for a comparison between neural recordings and spiking in our model). In

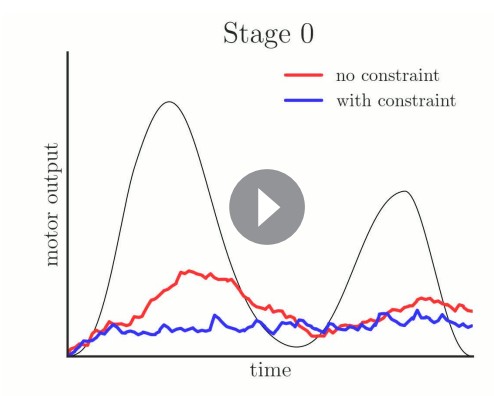

**Video 3.** Effects of adding a constraint on tutor firing rates on the evolution of motor output during learning in a rate-based simulation. The plasticity rule here was rate-based ($\alpha = 0$). This video relates to *Figure 4A*.

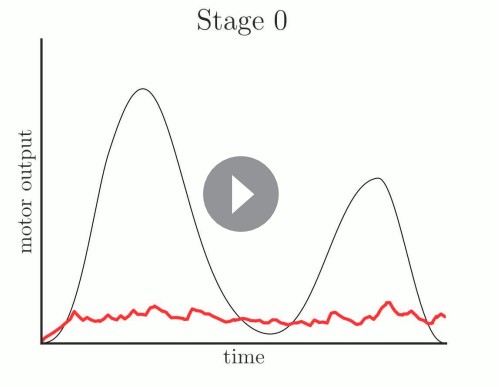

**Video 4.** Evolution of motor output showing sequential learning in a rate-based simulation when the firing rate constraint is imposed on a tutor with a long memory timescale. This video relates to *Figure 4B*.

contrast to the constant inhibitory bias that we used in our rate-based simulations, for the spiking simulations we chose an activity-dependent global inhibition for RA neurons. We also tested that a constant bias produced similar results (see Appendix).

Synaptic strength updates followed the same two-timescale dynamics that was used in the rate-based models (*Figure 2B*). The firing rates $c_i(t)$ and $g_j(t)$ that appear in the plasticity equation were calculated in the spiking model by filtering the spike trains from conductor and tutor neurons with exponential kernels. The synaptic weights were constrained to be non-negative. (See Materials and methods for details.)

As long as the tutor error integration timescale was not too large, learning proceeded effectively when the tutor error integration timescale and the student plasticity rule were matched (see *Figure 5C* and online *Video 5*), with mismatches slowing down or abolishing learning, just as in our rate-based study (compare *Figure 5D* with *Figure 3C*). The rate of learning and the accuracy of the trained state were lower in the spiking model compared to the rate-based model. The lower accuracy arises because the tutor neurons fire stochastically, unlike the deterministic neurons used in the rate-based simulations. The stochastic nature of the tutor firing also led to a decrease in learning accuracy as the tutor error integration timescale $\tau_{\text{tutor}}$ increased (*Figure 5D*). This happens through two related effects: (1) the signal-to-noise ratio in the tutor guiding signal decreases as $\tau_{\text{tutor}}$ increases once the tutor error integration timescale is longer than the duration $T$ of the motor program (see Appendix); and (2) the fluctuations in the conductor–student weights lead to some weights getting clamped at 0 due to the positivity constraint, which leads to the motor program overshooting the target (see Appendix). The latter effect can be reduced by either allowing for negative weights, or changing the motor output to a push-pull architecture in which some student neurons enhance the output while others inhibit it. The signal-to-noise ratio effect can be attenuated by increasing the gain of the tutor signal, which inhibits early learning, but improves the quality of the guiding signal in the latter stages of the learning process. It is also worth emphasizing that these effects only become relevant once the tutor error integration timescale $\tau_{\text{tutor}}$ becomes significantly longer than the duration of the motor program, $T$, which for a birdsong motif would be around 1 s.

Spiking in our model tends to be a little more regular than that in the recordings (compare *Figure 5A* and *Figure 5B*). This could be due to sources of noise that are present in the brain which we did not model. One detail that our model does not capture is the fact that many LMAN spikes occur in bursts, while in our simulation LMAN firing is Poisson. Bursts are more likely to produce spikes in downstream RA neurons particularly because of the NMDA dynamics, and thus a bursty LMAN will be more effective at injecting variability into RA (*Kojima et al., 2013*). Small inaccuracies in aligning the recorded spikes to the song are also likely to contribute apparent variability between renditions in experiments. Indeed, some of the variability in *Figure 5A* looks like it could be due to time warping and global time shifts that were not fully corrected.

## Robust learning with credit assignment errors

The calculation of the tutor output in our rule involved estimating the motor error $\epsilon_j$ from *Equation (2)*. This required knowledge of the assignment between student activities and motor output, which in our model was represented by the matrix $M_{aj}$ (*Figure 2A*). In our simulations, we typically chose an assignment in which each student neuron contributed to a single output channel, mimicking the empirical findings for neurons in bird RA. Mathematically, this implies that each column of $M_{aj}$ contained a single non-zero element. In *Figure 6A*, we show what happened in the rate-based model when the tutor incorrectly assigned a certain fraction of the neurons to the wrong output. Specifically, we considered two output channels, $y_1$ and $y_2$, with half of the student neurons contributing only to $y_1$ and the other half contributing only to $y_2$. We then scrambled a fraction $\rho$ of this assignment when calculating the motor error, so that the tutor effectively had an imperfect knowledge of the student–output relation. *Figure 6A* shows that learning is robust to this kind of mis-assignment even for fairly large values of the error fraction $\rho$ up to about 40%, but quickly deteriorates as this fraction approaches 50%.

Due to environmental factors that affect development of different individuals in different ways, it is unlikely that the student–output mapping can be innate. As such, the tutor circuit must learn the mapping. Indeed, it is known that LMAN in the bird receives an indirect evaluation signal *via* Area X, which might be used to effect this learning (*Andalman and Fee, 2009*; *Gadagkar et al., 2016*;

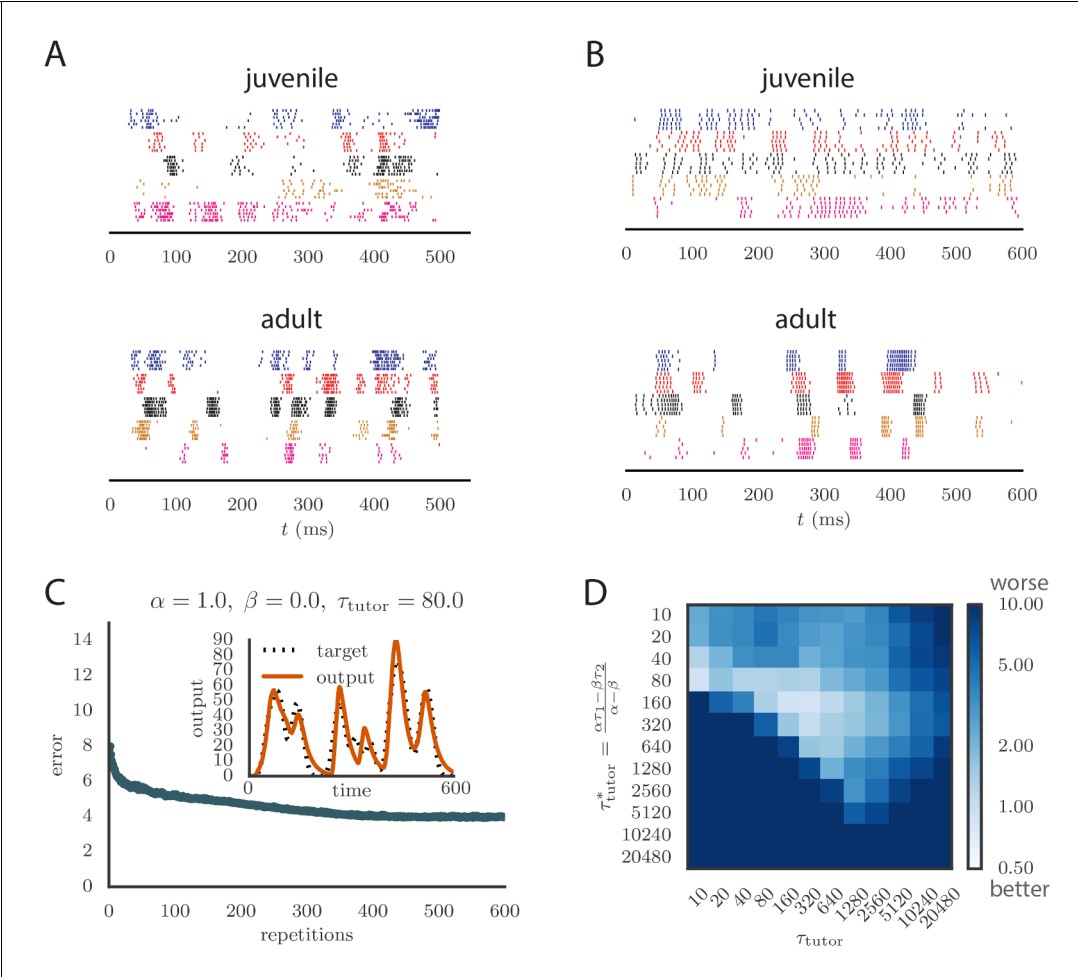

**Figure 5.** Results from simulations in spiking neural networks. (**A**) Spike patterns recorded from zebra finch RA during song production, for a juvenile (top) and an adult (bottom). Each color corresponds to a single neuron, and the song-aligned spikes for six renditions of the song are shown. Adapted from *Ölveczky et al. (2011)*. (**B**) Spike patterns from model student neurons in our simulations, for the untrained (top) and trained (bottom) models. The training used $\alpha = 1$, $\beta = 0$, and $\tau_{\text{tutor}} = 80$ ms, and ran for 600 iterations of the song. Each model neuron corresponds to a different output channel of the simulation. In this case, the targets for each channel were chosen to roughly approximate the time course observed in the neural recordings. (**C**) Progression of reproduction error in the spiking simulation as a function of the number of repetitions for the same conditions as in panel B. The inset shows the accuracy of reproduction in the trained model for one of the output channels. (See online *Video 5*.) (**D**) Effects of mismatch between student and tutor on reproduction accuracy in the spiking model. The heatmap shows the final reproduction error of the motor output after 1000 learning cycles in a spiking simulation where a student with parameters $\alpha$, $\beta$, $\tau_1$, and $\tau_2$ was paired with a tutor with memory timescale $\tau_{\text{tutor}}$. On the y axis, $\tau_1$ and $\tau_2$ were kept fixed at 80 ms and 40 ms, respectively, while $\alpha$ and $\beta$ were varied (subject to the constraint $\alpha - \beta = 1$; see section "Learning in a rate-based model"). Different choices of $\alpha$ and $\beta$ lead to different optimal timescales $\tau_{\text{tutor}}^*$ according to *Equation (4)*. The diagonal elements correspond to matched tutor and student, $\tau_{\text{tutor}} = \tau_{\text{tutor}}^*$. Note that the color scale is logarithmic.

*Hoffmann et al., 2016*; *Kubikova et al., 2010*). One way in which this can be achieved is through a reinforcement paradigm. We thus considered a learning rule where the tutor circuit receives a reward signal that enables it to infer the student–output mapping. In general the output of the tutor circuit should depend on an integral of the motor error, as in *Equation (3)*, to best instruct the student. For simplicity, we start with the memory-less case, $\tau_{\text{tutor}} = 0$, in which only the instantaneous value of the motor error is reflected in the tutor signal; we then show how to generalize this for $\tau_{\text{tutor}} > 0$.

As before, we took the tutor neurons to fire Poisson spikes with time-dependent rates $f_j(t)$, which were initialized arbitrarily. Because of stochastic fluctuations, the actual tutor activity on any given

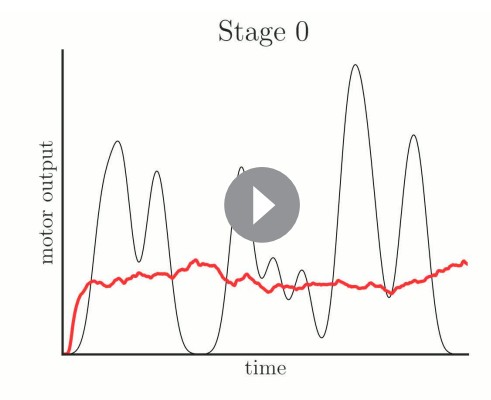

Stage 0

motor output

time

**Video 5.** Evolution of motor output during learning in a spiking simulation. The plasticity rule parameters were $\alpha = 1$, $\beta = 0$, and the tutor had a matching timescale $\tau_{\text{tutor}} = \tau_{\text{tutor}}^* = 80\,\text{ms}$. This video relates to *Figure 5C*.

trial, $g_j(t)$, differs somewhat from the average, $\bar{g}_j(t)$. Denoting the difference by $\xi_j(t) = g_j(t) - \bar{g}_j(t)$, the update rule for the tutor firing rates was given by

$$\Delta f_j(t) = \eta_{\text{tutor}}(R(t) - \bar{R})\xi_j(t)\,, \qquad (6)$$

where $\eta_{\text{tutor}}$ is a learning rate, $R(t)$ is the instantaneous reward signal, and $\bar{R}$ is its average over recent renditions of the motor program. In our implementation, $\bar{R}$ is obtained by convolving $R(t)$ with an exponential kernel (timescale = 1 s). The reward $R(t_{\text{max}})$ at the end of one rendition becomes the baseline at the start of the next rendition $R(0)$. The baseline $\bar{g}_j(t)$ of the tutor activity is calculated by averaging over recent renditions of the song with exponentially decaying weights (one *e*-fold of decay for every five renditions). Further implementation details are available in our code at https://github.com/ttesileanu/twostagelearning (*Teşileanu, 2016*) (with a copy archived at https://github.com/elifesciences-publications/twostagelearning).

The intuition behind this rule is that, whenever a fluctuation in the tutor activity leads to better-than-average reward ($R(t) > \bar{R}$), the tutor firing rate changes in the direction of the fluctuation for subsequent trials, 'freezing in' the improvement. Conversely, the firing rate moves away from the directions in which fluctuations tend to reduce the reward.

To test our learning rule, we ran simulations using this reinforcement strategy and found that learning again converges to an accurate rendition of the target output (*Figure 6B*, inset and online *Video 6*). The number of repetitions needed for training is greatly increased compared to the case in which the credit assignment is assumed known by the tutor circuit (compare *Figure 6B* to *Figure 5C*). This is because the tutor needs to use many training rounds for experimentation before it can guide conductor–student plasticity. The rate of learning in our model is similar to the songbird (*i.e.*, order 10 000 repetitions for learning, given that a zebra finch typically sings about 1000 repetitions of its song each day, and takes about one month to fully develop adult song).

Because of the extra training time needed for the tutor to adapt its signal, the motor output in our reward-based simulations tends to initially overshoot the target (leading to the kink in the error at around 2000 repetitions in *Figure 6B*). Interestingly, the subsequent reduction in output that leads to convergence of the motor program, combined with the positivity constraint on the synaptic strengths, leads to many conductor–student connections being pruned (*Figure 6D*). This mirrors experiments on songbirds, where the number of connections between HVC and RA first increases with learning and then decreases (*Garst-Orozco et al., 2014*).

The reinforcement rule described above responds only to instantaneous values of the reward signal and tutor firing rate fluctuations. In general, effective learning requires that the tutor keep a memory trace of its activity over a timescale $\tau_{\text{tutor}} > 0$, as in *Equation (4)*. To achieve this in the reinforcement paradigm, we can use a simple generalization of *Equation (6)* where the update rule is filtered over the tutor memory timescale:

$$\Delta f_j(t) = \eta_{\text{tutor}} \frac{1}{\tau_{\text{tutor}}} \int^t dt'\, (R(t') - \bar{R})\xi_j(t')e^{-(t-t')/\tau_{\text{tutor}}}\,. \qquad (7)$$

We tested that this rule leads to effective learning when paired with the corresponding student, *i.e.*, one for which *Equation (4)* is obeyed (*Figure 6C* and online *Video 7*).

The reinforcement rules proposed here are related to the learning rules from (*Fiete and Seung, 2006*; *Fiete et al., 2007*) and (*Farries and Fairhall, 2007*). However, those models focused on learning in a single pass, instead of the two-stage architecture that we studied. In particular, in *Fiete et al. (2007)*, area LMAN was assumed to generate pure Poisson noise and reinforcement

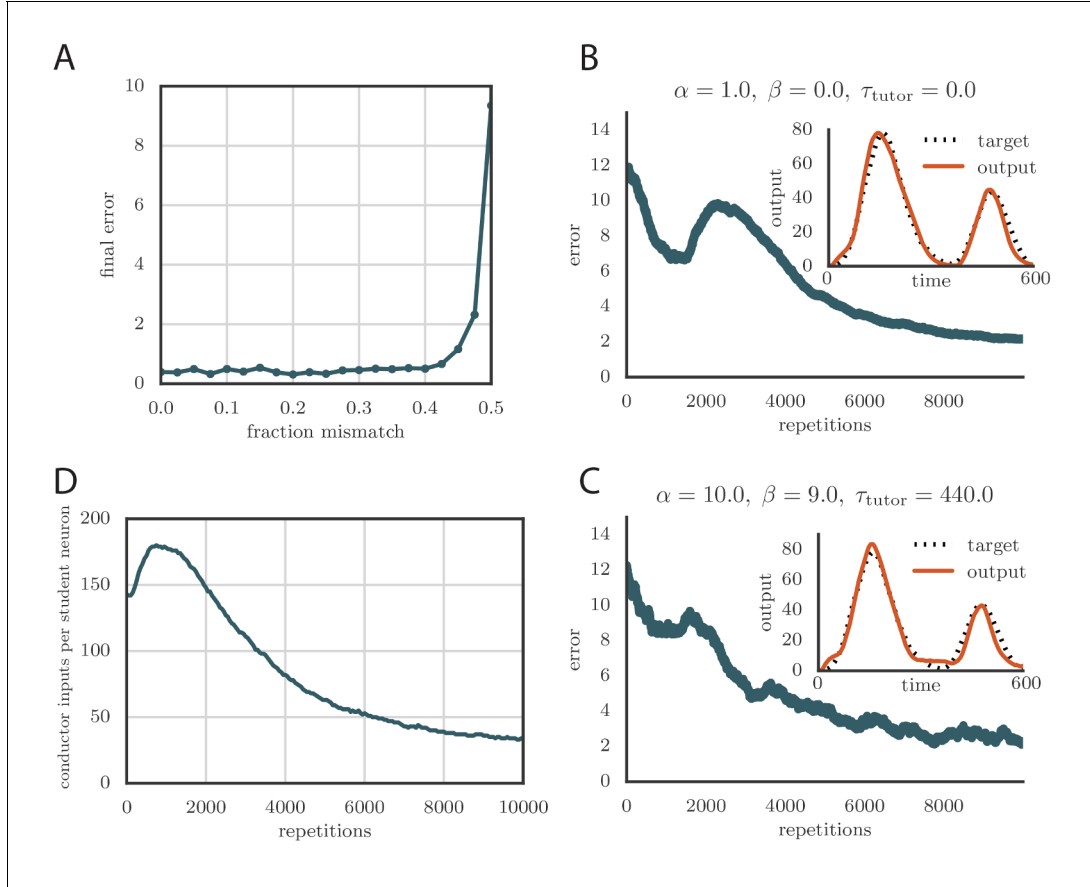

**Figure 6.** Credit assignment and reinforcement learning. (**A**) Effects of credit mis-assignment on learning in a rate-based simulation. Here, the system learned output sequences for two independent channels. The student–output weights $M_{aj}$ were chosen so that the tutor wrongly assigned a fraction of student neurons to an output channel different from the one it actually mapped to. The graph shows how the accuracy of the motor output after 1000 learning steps depended on the fraction of mis-assigned credit. (**B**) Learning curve and trained motor output (inset) for one of the channels showing two-stage reinforcement-based learning for the memory-less tutor ($\tau_{\text{tutor}} = 0$). The accuracy of the trained model is as good as in the case where the tutor was assumed to have a perfect model of the student–output relation. However, the speed of learning is reduced. (See online *Video 6*.) (**C**) Learning curve and trained motor output (inset) for one of the output channels showing two-stage reinforcement-based learning when the tutor circuit needs to integrate information about the motor error on a certain timescale. Again, learning was slow, but the accuracy of the trained state was unchanged. (See online *Video 7*.) (**D**) Evolution of the average number of HVC inputs per RA neuron with learning in a reinforcement example. Synapses were considered pruned if they admitted a current smaller than 1 nA after a pre-synaptic spike in our simulations.

learning took place at the HVC–RA synapses. In our model, which is in better agreement with recent evidence regarding the roles of RA and LMAN in birdsong (*Andalman and Fee, 2009*), reinforcement learning first takes place in the anterior forebrain pathway (AFP), for which LMAN is the output. A reward-independent heterosynaptic plasticity rule then solidifies the information in RA.

In our simulations, tutor neurons fire Poisson spikes with specific time-dependent rates which change during learning. The timecourse of the firing rates in each repetition must then be stored somewhere in the brain. In fact, in the songbird, there are indirect projections from HVC to LMAN, going through the basal ganglia (Area X) and the dorso-lateral division of the medial thalamus (DLM) in the anterior forebrain pathway (*Figure 1A*) (*Perkel, 2004*). These synapses could store the required time-dependence of the tutor firing rates. In addition, the same synapses can provide the timebase input that would ensure synchrony between LMAN firing and RA output, as necessary for learning. Our reinforcement learning rule for the tutor area, *Equation (6)*, can be viewed as an effective model for plasticity in the projections between HVC, Area X, DLM, and LMAN, as in *Fee and Goldberg (2011)*. In this picture, the indirect HVC–LMAN connections behave somewhat like the 'hedonistic synapses' from *Seung (2003)*, though we use a simpler synaptic model here.

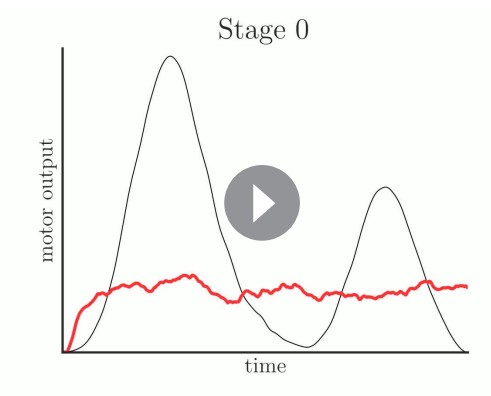

**Video 6.** Evolution of motor output during learning in a spiking simulation with a reinforcement-based tutor. Here the tutor was memory-less ($\tau_{\text{tutor}} = 0$). This video relates to *Figure 6B*.

Implementing the integral from *Equation (7)* would require further recurrent circuitry in LMAN which is beyond the scope of this paper, but would be interesting to investigate in future work.

## Discussion

We built a two-stage model of learning in which one area (the student) learns to generate a patterned motor output under guidance from a tutor area. This architecture is inspired by the song system of zebra finches, where area LMAN provides a corrective bias to the song that is then consolidated in the HVC–RA synapses. Using an approach rooted in the efficient coding literature, we showed analytically that, in a simple model, the tutor output that is most likely to lead to effective learning by the student involves an integral over the recent magnitude of the motor error. We found that efficiency requires that the timescale for this integral should be related to the synaptic plasticity rule used by the student. Using simulations, we tested our findings in more general settings. In particular, we demonstrated that tutor-student matching is important for learning in a spiking-neuron model constructed to reproduce spiking patterns similar to those measured in zebra finches. Learning in this model changes the spiking statistics of student neurons in realistic ways, for example, by producing more bursty, stereotyped firing events as learning progresses. Finally, we showed how the tutor can build its error-correcting signal by means of reinforcement learning.

If the birdsong system supports efficient learning, our model can predict the temporal structure of the firing patterns of RA-projecting LMAN neurons, given the plasticity rule implemented at the HVC–RA synapses. These predictions can be directly tested by recordings from LMAN neurons in singing birds, assuming that a good measure of motor error is available, and that we can estimate how the neurons contribute to this error. Moreover, recordings from a tutor circuit, such as LMAN, could be combined with a measure of motor error to infer the plasticity rule in a downstream student circuit, such as RA. This could be compared with direct measurements of the plasticity rule obtained in slice. Conversely, knowledge of the student plasticity rule could be used to predict the time-dependence of tutor firing rates. According to our model, the firing rate should reflect the integral of the motor error with the timescale predicted by the model. A different approach would be to artificially tutor RA by stimulating LMAN neurons electrically or optogenetically. We would predict that if the tutor signal is delivered appropriately (*e.g.*, in conjunction with a particular syllable [*Tumer and Brainard, 2007*]), then the premotor bias produced by the stimulation should become incorporated into the motor pathway faster when the timescale of the artificial LMAN signal is properly matched to the RA synaptic plasticity rule.

Our model can be applied more generally to other systems in the brain exhibiting two-stage learning, such as motor learning in mammals. If the plasticity mechanisms in these systems are different from those in songbirds, our

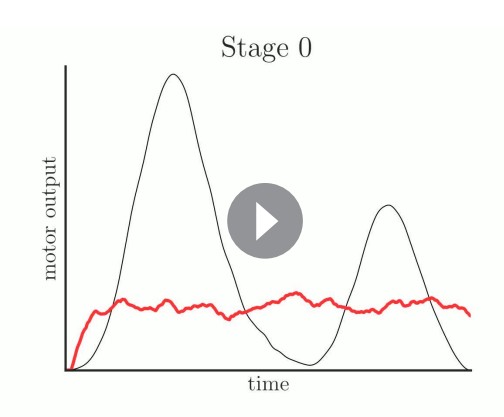

**Video 7.** Evolution of motor output during learning in a spiking simulation with a reinforcement-based tutor. Here the tutor needed to integrate information about the motor error on a timescale $\tau_{\text{tutor}} = 440\,\text{ms}$. This video relates to *Figure 6C*.

predictions for the structure of the guiding signal will vary correspondingly. This would allow a further test of our model of 'efficient learning' in the brain. It is worth pointing out that our model was derived assuming a certain hierarchy among the timescales that model the student plasticity and the tutor signal. A mismatch between the model predictions and observations could also imply a breakdown of these approximations, rather than failure of the hypothesis that the particular system under study evolved to support efficient learning. Of course our analysis could be extended by relaxing these assumptions, for example by keeping more terms in the Taylor expansion that we used in our derivation of the matched tutor signal.

Applied to birdsong, our model is best seen as a mechanism for learning song syllables. The ordering of syllables in song motifs seems to have a second level of control within HVC and perhaps beyond (*Basista et al., 2014*; *Hamaguchi et al., 2016*). Songs can also be distorted by warping their timebase through changes in HVC firing without alterations of the HVC–RA connectivity (*Ali et al., 2013*). In view of these phenomena, it would be interesting to incorporate our model into a larger hierarchical framework in which the sequencing and temporal structure of the syllables are also learned. A model of transitions between syllables can be found in *Doya and Sejnowski (2000)*, where the authors use a 'weight perturbation' optimization scheme in which each HVC–RA synaptic weight is perturbed individually. We did not follow this approach because there is no plausible mechanism for LMAN to provide separate guidance to each HVC–RA synapse; in particular, there are not enough LMAN neurons (*Fiete et al., 2007*).

In this paper we assumed a two-stage architecture for learning, inspired by birdsong. An interesting question is whether and under what conditions such an architecture is more effective than a single-step model. Possibly, having two stages is better when a single tutor area is responsible for training several different dedicated controllers, as is likely the case in motor learning. It would then be beneficial to have an area that can learn arbitrary behaviors, perhaps at the cost of using more resources and having slower reaction times, along with the ability to transfer these behaviors into low-level circuitry that is only capable of producing stereotyped motor programs. The question then arises whether having more than two levels in this hierarchy could be useful, what the other levels might do, and whether such hierarchical learning systems are implemented in the brain.

## Materials and methods

### Equations for rate-based model

The basic equations we used for describing our rate-based model (*Figure 2A*) are the following:

$$
\begin{aligned}
y_a(t) &= \sum_j M_{aj} s_j(t), \\
s_j(t) &= \sum_i W_{ij} c_i(t) + w g_j(t) - x_{\text{inh}}.
\end{aligned}
\tag{8}
$$

In simulations, we further filtered the output using an exponential kernel,

$$
\tilde{y}_a(t) = \sum_j M_{aj} \int_0^t s_j(t') e^{-(t-t')/\tau_{\text{out}}} \, dt',
\tag{9}
$$

with a timescale $\tau_{\text{out}}$ that we typically set to 25 ms. The smoothing produces more realistic outputs by mimicking the relatively slow reaction time of real muscles, and stabilizes learning by filtering out high-frequency components of the motor output. The latter interfere with learning because of the delay between the effect of conductor activity on synaptic strengths *vs.* motor output. This delay is of the order $\tau_{1,2} - \tau_{\text{out}}$ (see the plasticity rule below).

The conductor activity in the rate-based model is modeled after songbird HVC (*Hahnloser et al., 2002*): each neuron fires a single burst during the motor program. Each burst corresponds to a sharp increase of the firing rate $c_i(t)$ from 0 to a constant value, and then a decrease $10\,\text{ms}$ later. The activities of the different neurons are spread out to tile the whole duration of the output program. Other choices for the conductor activity also work, provided no patterns are repeated (see Appendix).

## Mathematical description of plasticity rule

In our model the rate of change of the synaptic weights obeys a rule that depends on a filtered version of the conductor signal (see *Figure 2B*). This is expressed mathematically as

$$\frac{dW_{ij}}{dt} = \eta \, \tilde{c}_i(t) \, (g_j(t) - \theta), \tag{10}$$

where $\eta$ is a learning rate and $\tilde{c}_i = K * c_i$, with the star representing convolution and $K$ being a filtering kernel. We considered a linear combination of two exponential kernels with timescales $\tau_1$ and $\tau_2$,

$$K(t) = \alpha K_1(t) - \beta K_2(t), \tag{11}$$

with $K_i(t)$ given by

$$K_i(t) = \begin{cases} \tau_i^{-1} e^{-t/\tau_i} & \text{for } t \geq 0, \\ 0 & \text{else.} \end{cases} \tag{12}$$

Different choices for the kernels give similar results (see Appendix). The overall scale of $\alpha$ and $\beta$ can be absorbed into the learning rate $\eta$ in *Equation (10)*. In our simulations, we fix $\alpha - \beta = 1$ and keep the learning rate constant as we change the plasticity rule (see *Equation 3*).

In the spiking simulations with and without reinforcement learning in the tutor circuit, the firing rates $c_i(t)$ and $g_j(t)$ were estimated by filtering spike trains with exponential kernels whose timescales were in the range $5\,\text{ms}$–$40\,\text{ms}$. The reinforcement studies typically required longer timescales for stability, possibly because of delays between conductor activity and reward signals.

## Derivation of the matching tutor signal

To find the tutor signal that provides the most effective teaching for the student, we first calculate how much synaptic weights change according to our plasticity rule, *Equation (10)*. Then we require that this change matches the gradient descent direction. We have

$$\Delta W_{ij} = \int_0^T \frac{dW_{ij}}{dt} \, dt = \eta \int_0^T \tilde{c}_i(t)(g_j(t) - \theta) \, dt. \tag{13}$$

Because of the linearity assumptions in our model, it is sufficient to focus on a case in which each conductor neuron, $i$, fires a single short burst, at a time $t_i$. We write this as $c_i(t) = \delta(t - t_i)$, and so

$$\Delta W_{ij} = \int_0^T \frac{dW_{ij}}{dt} \, dt = \eta \int_0^T K(t - t_i)(g_j(t) - \theta) \, dt, \tag{14}$$

where we used the definition of $\tilde{c}_i(t)$. If the time constants $\tau_1$, $\tau_2$ are short compared to the timescale on which the tutor input $g_j(t)$ varies, only the values of $g_j(t)$ around time $t_i$ will contribute to the integral. If we further assume that $T \gg t_i$, we can use a Taylor expansion of $g_j(t)$ around $t = t_i$ to perform the calculation:

$$\begin{aligned} \Delta W_{ij} &\approx \eta \int_{t_i}^{\infty} K(t - t_i)\big(g_j(t_i) - \theta + (t - t_i)g_j'(t_i)\big) \, dt \\ &= \eta(g_j(t_i) - \theta) \int_0^{\infty} K(t) \, dt + \eta g_j'(t_i) \int_0^{\infty} t K(t) \, dt \\ &= \eta(g_j(t_i) - \theta) \int_0^{\infty} \big(\alpha K_1(t) - \beta K_2(t)\big) \, dt + \eta g_j'(t_i) \int_0^{\infty} t \big(\alpha K_1(t) - \beta K_2(t)\big) \, dt. \end{aligned} \tag{15}$$

Doing the integrals involving the exponential kernels $K_1$ and $K_2$, we get

$$\Delta W_{ij} = \eta \big[(\alpha - \beta)(g_j(t_i) - \theta) + (\alpha \tau_1 - \beta \tau_2) g_j'(t_i)\big]. \tag{16}$$

We would like this synaptic change to optimally reduce a measure of mismatch between the output and the desired target as measured by a loss function. A generic smooth loss function $L(y_a(t), \bar{y}_a(t))$ can be quadratically approximated when $y_a$ is sufficiently close to the target $\bar{y}_a(t)$. With this in mind, we consider a quadratic loss

$$L = \frac{1}{2} \sum_a \int_0^T \left[ y_a(t) - \bar{y}_a(t) \right]^2 dt. \tag{17}$$

The loss function would decrease monotonically during learning if synaptic weights changed in proportion to the negative gradient of $L$:

$$\Delta W_{ij} = -\gamma \frac{\partial L}{\partial W_{ij}}, \tag{18}$$

where $\gamma$ is a learning rate. This implies

$$\Delta W_{ij} = -\gamma \sum_a \int_0^T M_{aj} \left[ y_a(t) - \bar{y}_a(t) \right] c_i(t). \tag{19}$$

Using again $c_i(t) = \delta(t - t_i)$, we obtain

$$\Delta W_{ij} = -\gamma \epsilon_j(t_i), \tag{20}$$

where we used the notation from *Equation (2)* for the motor error at student neuron $j$.

We now set *Equations (16) and (20)* equal to each other. If the conductor fires densely in time, we need the equality to hold for all times, and we thus get a differential equation for the tutor signal $g_j(t)$. This identifies the tutor signal that leads to gradient descent learning as a function of the motor error $\epsilon_j(t)$, *Equation (3)* (with the notation $\zeta = \gamma/\eta$).

## Spiking simulations

We used spiking models that were based on leaky integrate-and-fire neurons with current-based dynamics for the synaptic inputs. The magnitude of synaptic potentials generated by the conductor–student synapses was independent of the membrane potential, approximating AMPA receptor dynamics, while the synaptic inputs from the tutor to the student were based on a mixture of AMPA and NMDA dynamics. Specifically, the equations describing the dynamics of the spiking model were:

$$
\begin{aligned}
\tau_m \frac{dV_j}{dt} &= (V_R - V_j) + R \left( I_j^{\text{AMPA}} + I_j^{\text{NMDA}} \right) - V_{\text{inh}}, \quad \text{(except during refractory period)} \\
\frac{dI_j^{\text{AMPA}}}{dt} &= -\frac{I_j^{\text{AMPA}}}{\tau_{\text{AMPA}}} + \sum_i W_{ij} \sum_k \delta(t - t_k^{\text{conductor}\#i}) + (1-r)w \sum_k \delta(t - t_k^{\text{tutor}}), \\
\frac{dI_j^{\text{NMDA}}}{dt} &= -\frac{I_j^{\text{NMDA}}}{\tau_{\text{NMDA}}} + rwG(V_j) \sum_k \delta(t - t_k^{\text{tutor}}), \\
V_{\text{inh}} &= \frac{g_{\text{inh}}}{N_{\text{student}}} \sum_j S_j(t), \\
\frac{dS_j}{dt} &= -\frac{S_j}{\tau_{\text{inh}}} + \sum_k \delta(t - t_k^{\text{student}}), \\
G(V) &= \left[ 1 + \frac{[\text{Mg}]}{3.57\,\text{mM}} \exp(-V/16.13\,\text{mV}) \right]^{-1}.
\end{aligned}
\tag{21}
$$

Here $V_j$ is the membrane potential of the $j^{\text{th}}$ student neuron and $V_R$ is the resting potential, as well as the potential to which the membrane was reset after a spike. Spikes were registered whenever the membrane potential went above a threshold $V_{\text{th}}$, after which a refractory period $\tau_{\text{ref}}$ ensued. Apart from excitatory AMPA and NMDA inputs modeled by the $I_j^{\text{AMPA}}$ and $I_j^{\text{NMDA}}$ variables in our model, we also included a global inhibitory signal $V_{\text{inh}}$ which is proportional to the overall activity of student neurons averaged over a timescale $\tau_{\text{inh}}$. The averaging is performed using the auxiliary variables $S_j$ which are convolutions of student spike trains with an exponential kernel. These can be thought of as a simple model for the activities of inhibitory interneurons in the student.

*Table 1* gives the values of the parameters we used in the simulations. These values were chosen to match the firing statistics of neurons in bird RA, as described below.

The voltage dynamics for conductor and tutor neurons was not simulated explicitly. Instead, each conductor neuron was assumed to fire a burst at a fixed time during the simulation. The onset of

**Table 1.** Values for parameters used in the spiking simulations.

| Parameter | Symbol | Value | Parameter | Symbol | Value |
|---|---|---|---|---|---|
| No. of conductor neurons | | 300 | No. of student neurons | | 80 |
| Reset potential | $V_R$ | $-72.3\,\mathrm{mV}$ | Input resistance | $R$ | $353\,\mathrm{M\Omega}$ |
| Threshold potential | $V_{\mathrm{th}}$ | $-48.6\,\mathrm{mV}$ | Strength of inhibition | $g_{\mathrm{inh}}$ | $1.80\,\mathrm{mV}$ |
| Membrane time constant | $\tau_m$ | $24.5\,\mathrm{ms}$ | Fraction NMDA receptors | $r$ | 0.9 |
| Refractory period | $\tau_{\mathrm{ref}}$ | $1.1\,\mathrm{ms}$ | Strength of synapses from tutor | $w$ | $100\,\mathrm{nA}$ |
| AMPA time constant | $\tau_{\mathrm{AMPA}}$ | $6.3\,\mathrm{ms}$ | No. of conductor synapses per student neuron | | 148 |
| NMDA time constant | $\tau_{\mathrm{NMDA}}$ | $81.5\,\mathrm{ms}$ | Mean strength of synapses from conductor | | $32.6\,\mathrm{nA}$ |
| Time constant for global inhibition | $\tau_{\mathrm{inh}}$ | $20\,\mathrm{ms}$ | Standard deviation of conductor–student weights | | $17.4\,\mathrm{nA}$ |
| Conductor firing rate during bursts | | $632\,\mathrm{Hz}$ | | | |

each burst had additive timing jitter of $\pm 0.3\,\mathrm{ms}$ and each spike in the burst had a jitter of $\pm 0.2\,\mathrm{ms}$. This modeled the uncertainty in spike times that is observed in *in vivo* recordings in birdsong (*Hahnloser et al., 2002*). Tutor neurons fired Poisson spikes with a time-dependent firing rate that was set as described in the main text.

The initial connectivity between conductor and student neurons was chosen to be sparse (see *Table 1*). The initial distribution of synaptic weights was log-normal, matching experimentally measured values for zebra finches (*Garst-Orozco et al., 2014*). Since these measurements are done in the slice, the absolute number of HVC synapses per RA neuron is likely to have been underestimated. The number of conductor–student synapses we start with in our simulations is thus chosen to be higher than the value reported in that paper (see *Table 1*), and is allowed to change during learning. We checked that the learning paradigm described here is robust to substantial changes in these parameters, but we have chosen values that are faithful to birdsong experiments and which are thus able to imitate the RA spiking statistics during song.

The synapses projecting onto each student neuron from the tutor have a weight that is fixed during our simulations reflecting the finding in *Garst-Orozco et al. (2014)* that the average strength of LMAN–RA synapses for zebra finches does not change with age. There is some evidence that individual LMAN–RA synapses undergo plasticity concurrently with the HVC–RA synapses (*Mehaffey and Doupe, 2015*) but we did not seek to model this effect. There are also developmental changes in the kinetics of NMDA-mediated synaptic currents in both HVC–RA and LMAN–RA synapses which we do not model (*Stark and Perkel, 1999*). These, however, happen early in development, and thus are unlikely to have an effect on song crystallization, which is what our model focuses on. *Stark and Perkel, 1999* also observed changes in the relative contribution of NMDA to AMPA responses in the HVC–RA synapses. We do not incorporate such effects in our model since we do not explicitly model the dynamics of HVC neurons in this paper. However, this is an interesting avenue for future work, especially since there is evidence that area HVC can also contribute to learning, in particular in relation to the temporal structure of song (*Ali et al., 2013*).

## Matching spiking statistics with experimental data

We used an optimization technique to choose parameters to maximize the similarity between the statistics of spiking in our simulations and the firing statistics observed in neural recordings from the songbird. The comparison was based on several descriptive statistics: the average firing rate; the coefficient of variation and skewness of the distribution of inter-spike intervals; the frequency and average duration of bursts; and the firing rate during bursts. For calculating these statistics, bursts were defined to start if the firing rate went above 80 Hz and last until the rate decreased below 40 Hz.

To carry out such optimizations in the stochastic context of our simulations, we used an evolutionary algorithm—the covariance matrix adaptation evolution strategy (CMA-ES) (*Hansen, 2006*). The objective function was based on the relative error between the simulation statistics $x_i^{\mathrm{sim}}$ and the observed statistics $x_i^{\mathrm{obs}}$,

$$\text{error} = \left[ \sum_i \left( \frac{x_i^{\text{sim}}}{x_i^{\text{obs}}} - 1 \right)^2 \right]^{1/2}.  \qquad (22)$$

Equal weight was placed on optimizing the firing statistics in the juvenile (based on a recording from a 43 dph bird) and optimizing firing in the adult (based on a recording from a 160 dph bird). In this optimization there was no learning between the juvenile and adult stages. We simply required that the number of HVC synapses per RA neuron, and the mean and standard deviation of the corresponding synaptic weights were in the ranges seen in the juvenile and adult by *Garst-Orozco et al. (2014)*. The optimization was carried out in Python (RRID:SCR_008394), using code from https://www.lri.fr/~hansen/cmaes_inmatlab.html. The results fixed the parameter choices in *Table 1* which were then used to study our learning paradigm. While these choices are important for achieving firing statistics that are similar to those seen in recordings from the bird, our learning paradigm is robust to large variations in the parameters in *Table 1*.

### Software and data
We used custom-built Python (RRID:SCR_008394) code for simulations and data analysis. The software and data that we used can be accessed online on GitHub (RRID:SCR_002630) at https://github.com/ttesileanu/twostagelearning.

## Acknowledgements
We would like to thank Serena Bradde for fruitful discussions during the early stages of this work. We also thank Xuexin Wei and Christopher Glaze for useful discussions. We are grateful to Timothy Otchy for providing us with some of the data we used in this paper. During this work VB was supported by NSF grant PHY-1066293 at the Aspen Center for Physics and by NSF Physics of Living Systems grant PHY-1058202. TT was supported by the Swartz Foundation.

## Additional information

### Funding

| Funder | Author |
| --- | --- |
| Swartz Foundation | Tiberiu Teşileanu |
| National Science Foundation | Vijay Balasubramanian |

The funders had no role in study design, data collection and interpretation, or the decision to submit the work for publication.

### Author contributions
TT, Conceptualization, Software, Formal analysis, Validation, Investigation, Visualization, Methodology, Writing—original draft, Writing—review and editing; BÖ, Conceptualization, Supervision, Investigation, Methodology, Writing—original draft, Writing—review and editing; VB, Conceptualization, Formal analysis, Supervision, Funding acquisition, Investigation, Writing—original draft, Writing—review and editing

### Author ORCIDs
Tiberiu Teşileanu, http://orcid.org/0000-0003-3107-3088
Bence Ölveczky, http://orcid.org/0000-0003-2499-2705
Vijay Balasubramanian, http://orcid.org/0000-0002-6497-3819

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

## Appendix 1

### Effect of nonlinearities

We can generalize the model from *Equation (8)* by using a nonlinear transfer function from student activities to motor output, and a nonlinear activation function for student neurons:

$$
\begin{aligned}
y_a(t) &= N_a\Big(\sum_j M_{aj} s_j(t)\Big),\\
s_j(t) &= F\Big(\sum_i W_{ij} c_i(t) + w g_j(t) - x_{\text{inh}}\Big).
\end{aligned}
\tag{23}
$$

Suppose further that we use a general loss function,

$$
L = \int_0^T \mathcal{L}\big(\{y_a(t) - \bar{y}_a(t)\}\big)\, dt.
\tag{24}
$$

Following the same argument as that from section "Derivation of the matching tutor signal", the gradient descent condition, *Equation (18)*, implies

$$
\Delta W_{ij} = -\gamma \int_0^T \sum_a M_{aj} N_a' F' c_i(t) \frac{\partial \mathcal{L}}{\partial y_a}\bigg|_{y_a(t) - \bar{y}_a(t)}.
\tag{25}
$$

The departure from the quadratic loss function, $\mathcal{L} \neq \frac{1}{2}\sum_a (y_a(t) - \bar{y}_a(t))^2$, and the nonlinearities in the output, $N_a$, have the effect of redefining the motor error,

$$
\epsilon_j(t) = \sum_a M_{aj} N_a' \frac{\partial \mathcal{L}}{\partial y_a}\bigg|_{y_a(t) - \bar{y}_a(t)}.
\tag{26}
$$

A proper loss function will be such that the derivatives $\partial \mathcal{L}/\partial y_a$ vanish when $y_a(t) = \bar{y}_a(t)$, and so the motor error $\epsilon_j$ as defined here is zero when the rendition is perfect, as expected. If we use a tutor that ignores the nonlinearities in a nonlinear system, *i.e.*, if we use *Equation (2)* instead of *Equation (26)* to calculate the tutor signal that is plugged into *Equation (3)*, we still expect successful learning provided that $N_a' > 0$ and that $\mathcal{L}$ is itself an increasing function of $|y_a - \bar{y}_a|$ (see section "Effect of different output functions"). This is because replacing *Equation (26)* with *Equation (2)* would affect the magnitude of the motor error without significantly changing its direction. In more complicated scenarios, if the transfer function to the output is not monotonic, there is the potential that using *Equation (2)* would push the system away from convergence instead of towards it. In such a case, an adaptive mechanism, such as the reinforcement rules from *Equations (6) or (7)* can be used to adapt to the local values of the derivatives $N_a'$ and $\partial \mathcal{L}/\partial y_a$.

Finally, the nonlinear activation function $F$ introduces a dependence on the student output $s_j(t)$ in *Equation (25)*, since $F'$ is evaluated at $F^{-1}(s_j(t))$. To obtain a good match between the student and the tutor in this context, we can modify the student plasticity rule (*Equation 10*) by adding a dependence on the postsynaptic activity,

$$
\frac{dW_{ij}}{dt} = \eta \tilde{c}_i(t)\, (g_j(t) - \theta)\, F'(F^{-1}(s_j(t))).
\tag{27}
$$

In general, synaptic plasticity has been observed to indeed depend on postsynaptic activity (*Chistiakova et al., 2014*; *Chistiakova and Volgushev, 2009*). Our derivation suggests that the effectiveness of learning could be improved by tuning this dependence of synaptic

change on postsynaptic activity to the activation function of postsynaptic neurons, according to *Equation (27)*. It would be interesting to check whether such tuning occurs in real neurons.

## Effect of different output functions

In the main text, we assumed a linear mapping between student activities and motor output. Moreover, we assumed a myotopic organization, in which each student neuron projected to a single muscle, leading to a student–output assignment matrix $M_{aj}$ in which each column had a single non-zero entry. We also assumed that student neurons only contributed additively to the outputs, with no inhibitory activity. Here we show that our results hold for other choices of student–output mappings.

For example, assume a push-pull architecture, in which half of the student neurons controlling one output are excitatory and half are inhibitory. This can be used to decouple the overall firing rate in the student from the magnitude of the outputs. Learning works just as effectively as in the case of the purely additive student–output mapping when using matched tutors, *Appendix 1—figures 1A and 1B*. The consequences of mismatching student and tutor circuits are also not significantly changed, *Appendix 1—figures 1C and 1D*.

We can also consider nonlinear mappings between the student activity and the final output. If there is a monotonic output nonlinearity, as in *Equation (23)* with $N'_a > 0$, the tutor signal derived for the linear case, *Equation (3)*, can still achieve convergence, though at a slower rate and with a somewhat lower accuracy (see *Appendix 1—figure 1E* for the case of a sigmoidal nonlinearity). For non-monotonic nonlinearities, the direction from which the optimum is approached can be crucial, as learning can get stuck in local minima of the loss function (we thank Josh Gold for this observation). Studying this might provide an interesting avenue to test whether learning in songbirds is based on a gradient descent-type rule or on a more sophisticated optimization technique.

Teşileanu *et al.* eLife 2017;6:e20944. DOI: 10.7554/eLife.20944

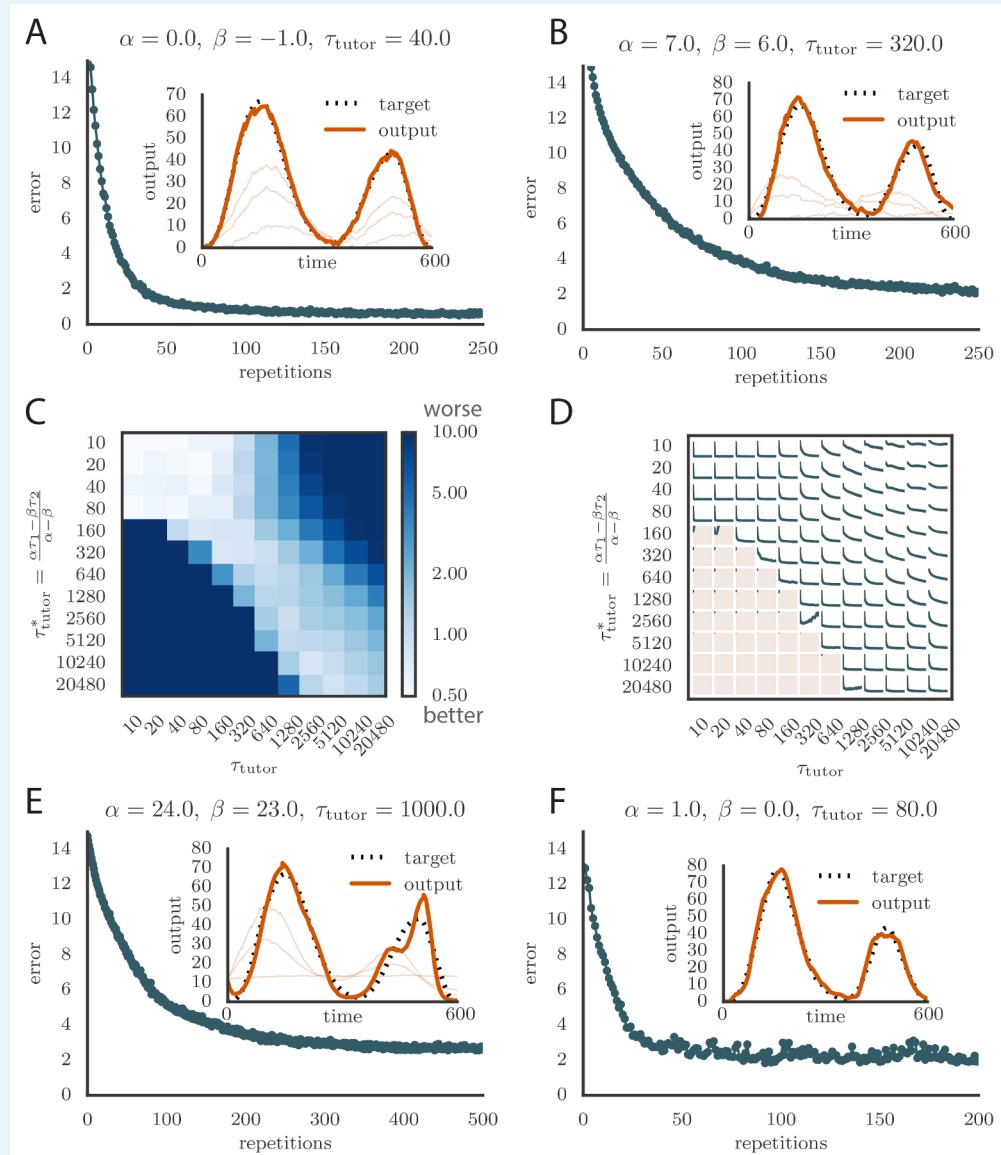

**Appendix 1—figure 1.** Robustness of learning. (**A**) Error trace showing how average motor error evolves with repetitions of the motor program for rate-based plasticity paired with a matching tutor, when the student–output mapping has a push-pull architecture. The inset shows the final motor output (thick red line) compared to the target output (dotted black line). The output on the first rendition and at two other stages of learning are also shown. (**B**) The error trace and final motor output shown for timing-based plasticity matched by a tutor with a long integration timescale. (**C**) Effects of mismatch between student and tutor on reproduction accuracy when using a push-pull architecture for the student–output mapping. The heatmap shows the final reproduction error of the motor output after 1000 learning cycles when a student with plasticity parameters $\alpha$ and $\beta$ is paired with a tutor with memory timescale $\tau_{\text{tutor}}$. Here $\tau_1 = 80\,\text{ms}$ and $\tau_2 = 40\,\text{ms}$. (**D**) Error evolution curves as a function of the mismatch between student and tutor. Each plot shows how the error in the motor program changes during 1000 learning cycles for the same conditions as those shown in the heatmap. The region shaded in light pink shows simulations where the mismatch between student and tutor leads to a deteriorating instead of improving performance during learning. (**E**) Convergence in the rate-based model with a linear-nonlinear controller that uses a sigmoidal nonlinearity. (**F**) Convergence in the spiking model when inhibition is constant instead of activity-dependent ($V_{\text{inh}} = \text{constant}$).

## Different inhibition models

In the spiking model, we used an activity-dependent inhibitory signal that was proportional to the average student activity. Using a constant inhibition instead, $V_{\text{inh}} = \text{constant}$, does not significantly change the results: see **Appendix 1—figure 1F** for an example.

## Effect of changing plasticity kernels

In the main text, we used exponential kernels with $\tau_1 = 80\,\text{ms}$ and $\tau_2 = 40\,\text{ms}$ for the smoothing of the conductor signal that enters the synaptic plasticity rule, **Equation (10)**. We can generalize this in two ways: we can use different timescales $\tau_1$, $\tau_2$, or we can use a different functional form for the kernels. (Note that in the main text we showed the effects of varying the parameters $\alpha$ and $\beta$ in the plasticity rule, while the timescales $\tau_1$ and $\tau_2$ were kept fixed.)

The values for the timescales $\tau_{1,2}$ were chosen to roughly match the shape of the plasticity curve measured in slices of zebra finch RA (**Mehaffey and Doupe, 2015**) (see **Figure 1C and D**). The main predictions of our model, that learning is most effective when the tutor signal is matched to the student plasticity rule, and that large mismatches between tutor and student lead to impaired learning, hold well when the student timescales change: see **Appendix 1—figure 2A** for the case when $\tau_1 = 20\,\text{ms}$ and $\tau_2 = 10\,\text{ms}$. In the main text we saw that the negative effects of tutor–student mismatch diminish for timescales that are shorter than $\sim \tau_{1,2}$. In **Appendix 1—figure 2A**, the range of timescales where a precise matching is not essential becomes very small because the student timescales are short.

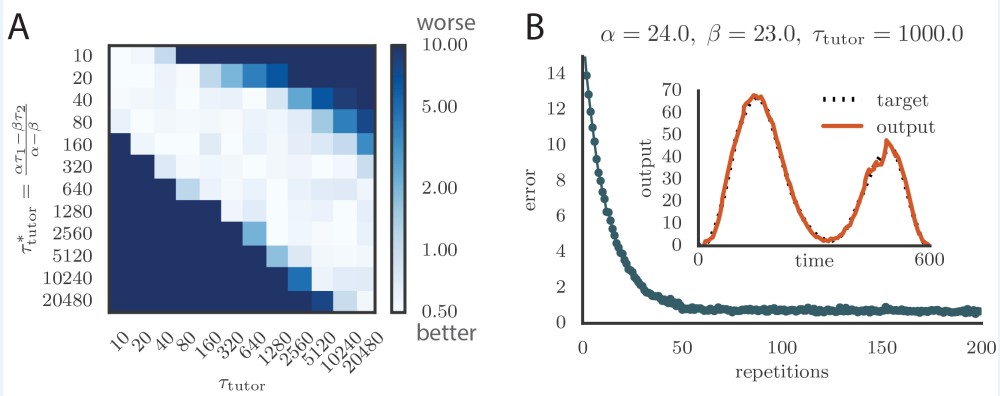

**Appendix 1—figure 2.** Effect of changing conductor smoothing kernels in the plasticity rule. (**A**) Matrix showing learning accuracy when using different timescales for the student plasticity rule. Each entry in the heatmap shows the average rendition error after 1000 learning steps when pairing a tutor with timescale $\tau_{\text{tutor}}$ with a non-matched student. Here the kernels are exponential, with timescales $\tau_1 = 20\,\text{ms}$, $\tau_2 = 10\,\text{ms}$. (**B**) Evolution of motor error with learning using kernels $\sim e^{-t/\tau}$ and $\sim te^{-t/\tau}$, instead of the two exponentials used in the main text. The tutor signal is as before, **Equation (3)**. The inset shows the final output for the trained model, for one of the two output channels. Learning is as effective and fast as before.

Another generalization of our plasticity rule can be obtained by changing the functional form of the kernels used to smooth the conductor input. As an example, suppose $K_2$ is kept exponential, while $K_1$ is replaced by

$$\bar{K}_1(t) = \begin{cases} \frac{1}{\bar{\tau}_1^2}\, t e^{-t/\bar{\tau}_1} & \text{for } t \geq 0, \\ 0 & \text{else.} \end{cases} \tag{28}$$

An example of learning using an STDP rule based on kernels $\bar{K}_1$ and $K_2$ where $\bar{\tau}_1 = \tau_2$ is shown in **Appendix 1—figure 2B**. The matching tutor has the same form as before, **Equation (3)** with timescale $\tau_{\text{tutor}} = \tau_{\text{tutor}}^*$ given by **Equation (4)**, but with $\tau_1 = 2\bar{\tau}_1 = 2\tau_2$. We can see that learning is as effective as in the case of purely exponential kernels.

## More general conductor patterns

In the main text, we have focused on a conductor whose activity matches that observed in area HVC of songbirds (**Hahnloser et al., 2002**): each neuron fires a single burst during the motor program. Our model, however, is not restricted to this case. We generated alternative conductor patterns by using arbitrarily-placed bursts of activity, as in **Appendix 1—figure 3A**. The model converges to a good rendition of the target program, **Appendix 1—figure 3B**. Learning is harder in this case because many conductor neurons can be active at the same time, and the weight updates affect not only the output of the system at the current position in the motor program, but also at all the other positions where the conductor neurons fire. This is in contrast to the HVC-like conductor, where each neuron fires at a single point in the motor program, and thus the effect of weight updates is better localized. More generally, simulations show that the sparser the conductor firing, the faster the convergence (data not shown). The accuracy of the final rendition of the motor program (**Appendix 1—figure 3B**, inset) is also not as good as before.

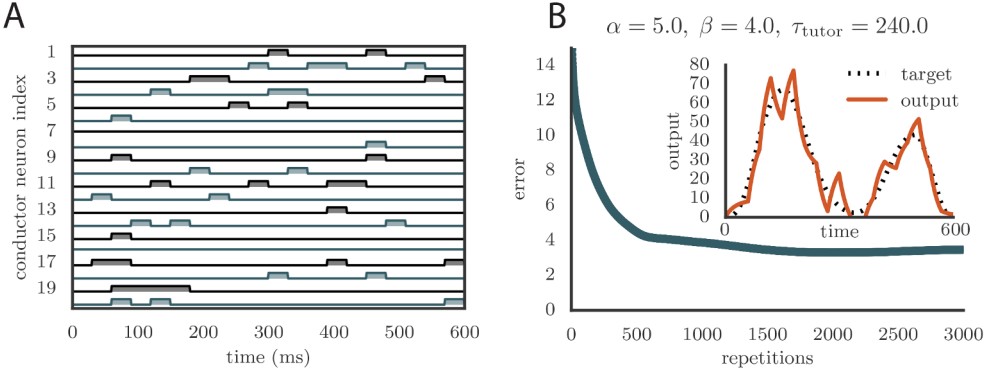

**Appendix 1—figure 3.** Learning with arbitrary conductor activity. (**A**). Typical activity of conductor neurons. 20 of the 100 neurons included in the simulation are shown. The activity pattern is chosen so that about 10% of the neurons are active at any given time. The pattern is chosen randomly but is fixed during learning. Each conductor burst lasts $30\,\text{ms}$. (**B**) Convergence curve and final rendition of the motor program (in inset). Learning included two output channels but the final output is shown for only one of them.

## Edge effects

In our derivation of the matching tutor rule, we assumed that the system has enough time to integrate all the synaptic weight changes from **Equation (10)**. However, some of these changes occur tens or hundreds of milliseconds after the inputs that generated them, due to the timescales used in the plasticity kernel. Since our simulations are only run for a finite amount of time, there will in general be edge effects, where periods of the motor program towards the end of the simulations will have difficulty converging. To offset such numerical

issues, we ran the simulations for a few hundred milliseconds longer than the duration of the motor program, and ignored the data from this extra period. Our simulations typically run for 600 ms, and the time reserved for relaxation after the end of the program was set to 1200 ms. The long relaxation time was chosen to allow for cases where the tutor was chosen to have a very long memory timescale.

## Parameter optimization for reproducing juvenile and adult spiking statistics

We set the parameters in our simulations to reproduce spiking statistics from recordings in zebra finch RA as closely as possible. *Appendix 1—figure 4* shows how the distribution of summary statistics obtained from 50 runs of the simulation compares to the distributions calculated from recordings in birds at various developmental stages. Each plot shows a standard box and whisker plot superimposed over a kernel-density estimate of the distribution of a given summary statistic, either over simulation runs or over recordings from birds at various stages of song learning. We ran two sets of simulations, one for a bird with juvenile-like connectivity between HVC and RA, and one with adult-like connectivity (see Materials and methods). In these simulations there was no learning to match the timecourse of songs—the goal was simply to identify parameters that lead to birdsong-like firing statistics.

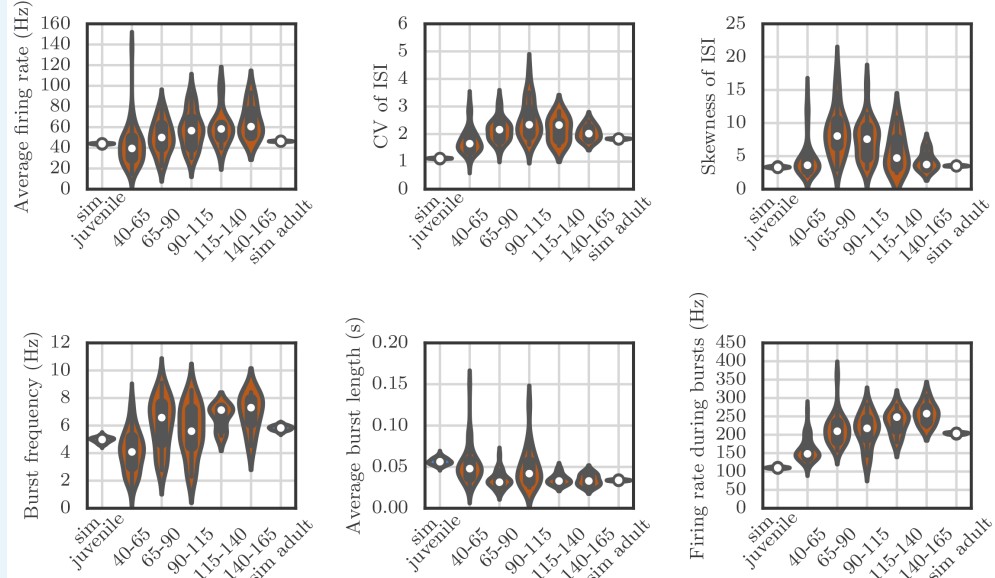

**Appendix 1—figure 4.** Violin plots showing how the spiking statistics from our simulation compared to the statistics obtained from neural recordings. Each violin shows a kernel-density estimate of the distribution that a particular summary statistic had in either several runs of a simulation, or in several recordings from behaving birds. The circle and the box within each violin show the median and the interquartile range.

The qualitative match between our simulations and recordings is good, but the simulations are less variable than the measurements. This may be due to sources of variability that we have ignored—for example, all our simulated neurons had exactly the same membrane time constants, refractory periods, and threshold potentials, which is not the case for real neurons. Another reason might be that in our simulations, all the runs were performed for the same network, while the measurements are from different cells in different birds.

## Effect of spiking stochasticity on learning

As pointed out in the main text, learning is affected in the spiking simulations when the tutor error integration timescale $\tau_{\text{tutor}}$ becomes very long. More specifically, two distinct effects occur. First, the fluctuations in the motor output increase, leading to a poorer match to the shape of the target motor program. And second, the whole output gets shifted up, towards higher muscle activation values. Both of these effects can be traced back to the stochasticity of the tutor signal.

In the spiking simulations, tutor neurons are assumed to fire Poisson spikes following a time-dependent firing rate that obeys *Equation (5)*. By the nature of the Poisson process, the tutor output in this case will contain fluctuations around the mean, $g(t) \sim \bar{g}(t) + \xi(t)$. Recall that the scale of $g(t)$ is set by the threshold $\theta$ and thus, since this is a Poisson process, so is the scale of the variability $\xi(t)$.

As long as the tutor error integration timescale is not very long, $g(t)$ roughly corresponds to a smoothed version of the motor error $\epsilon(t)$ (*cf. Equation 5*). However, as $\tau_{\text{tutor}}$ grows past the duration $T$ of the motor program, the exponential term in *Equation (5)* becomes essentially constant, leading to a tutor signal $\bar{g}(t)$ whose departures from the center value $\theta$ decrease in proportion to the timescale $\tau_{\text{tutor}}$. As far as the student is concerned, the relevant signal is $g(t) - \theta$ (*Equation 1*), and thus, when $\tau_{\text{tutor}} > T$, the signal-to-noise ratio in the tutor guiding signal starts to decrease as $1/\tau_{\text{tutor}}$. This ultimately leads to a very noisy rendition of the target program. One way to improve this would be to increase the gain factor $\zeta$ that controls the relation between the motor error and the tutor signal (see *Equation 5*). This improves the ability of the system to converge onto its target in the late stages of learning. In the early stages of learning, however, this could lead to saturation problems. One way to fix this would be to use a variable gain factor $\zeta$ that ensures the whole range of tutor firing rates is used without generating too much saturation. This would be an interesting avenue for future research.

Reducing the fluctuations in the tutor signal also decreases the fluctuations in the conductor–student synaptic weights, which leads to fewer weights being clamped at 0 because of the positivity constraint. This reduces the shift between the learned motor program and the target. As mentioned in the main text, another approach to reducing or eliminating this shift is to allow for negative weights or (more realistically) to use a push-pull mechanism, in which the activity of some student neurons acts to increase muscle output, while the activity of other student neurons acts as an inhibition on muscle output.

## Appendix 2

### Plasticity parameter values

In the heatmaps that appear in many of the figures in the main text and in the supplementary information, we kept the timescales $\tau_1$ and $\tau_2$ constant while varying $\alpha$ and $\beta$ to modify the student plasticity rule. Since the overall scale of $\alpha$ and $\beta$ is inconsequential as it can be absorbed into the learning rate (as explained in the section "Learning in a rate-based model"), we imposed the further constraint $\alpha - \beta = 1$. This implies that we effectively focused on a one-parameter family of student plasticity rule, as identified by the value of $\alpha$ (and the corresponding value for $\beta = \alpha - 1$). In the figures, we expressed this instead in terms of the timescale of the optimally-matching tutor, $\tau_{\text{tutor}}^*$, as defined in **Equation (4)**.

Below we give the explicit values of $\alpha$ and $\beta$ that we used for each row in the heatmaps. These can be calculated by solving for $\alpha$ in **Equation (4)**, using $\beta = \alpha - 1$, and assuming that $\tau_1 = 80\,\text{ms}$ and $\tau_2 = 40\,\text{ms}$.

| $\tau_{\text{tutor}}^*$ **(ms)** | $\alpha$ | $\beta$ |
| --- | --- | --- |
| 10 | −0.75 | −1.75 |
| 20 | −0.5 | −1.5 |
| 40 | 0.0 | −1.0 |
| 80 | 1.0 | 0.0 |
| 160 | 3.0 | 2.0 |
| 320 | 7.0 | 6.0 |
| 640 | 15.0 | 14.0 |
| 1280 | 31.0 | 30.0 |
| 2560 | 63.0 | 62.0 |
| 5120 | 127.0 | 126.0 |
| 10240 | 255.0 | 254.0 |
| 20480 | 511.0 | 510.0 |

