## [Decision Letter]

Thank you for submitting your article "Matching tutor to student: rules and mechanisms for efficient two-stage learning in neural circuits" for consideration by *eLife*. Your article has been reviewed by two peer reviewers, and the evaluation has been overseen by Michael Frank as the Reviewing Editor and Andrew King as the Senior Editor. The reviewers have opted to remain anonymous.

The reviewers have discussed the reviews with one another and the Reviewing Editor has drafted this decision to help you prepare a revised submission.

Summary:

This is a careful and well-designed study of the role of the matching between plasticity rules and instructive signals for efficient learning in a two-stage model, wherein the reinforcement signal is processed by one circuit that in turn provides input to another circuit that drives the actual behavior. This is particularly applicable to recent detailed findings in the birdsong field. Several important results are laid out in the manuscript: (1) matching is important for fast learning, (2) matching is not as important when the tutor timescale is shorter than STDP timescales (3) similar results are obtained for rate-based or spiking-based models, (4) learning drives student neurons to become more bursty, (5) learning drives pruning of connections in spiking networks, and (6) sequential learning is obtained when the tutor memory is long.

Essential revisions:

All involved found the work to be of high quality and of broad interest to researchers in sensorimotor and reinforcement learning, as well as being of particular interest to those working on neural recording in the songbird brain.

1) However, an invigorating consultation session among the reviewers was needed to get down to the bottom of whether some of the main modeling results were circular/trivial or not. It was determined that in fact they are not, and that rather some of them show that the model is robust to the precise form of the motor error. But the reason for this initial difference in opinion could be traced to a lack of clarity in the manuscript over the definition of tau_tutor. One reviewer originally understood it as the timescale over which the firing rate of the tutor neurons is modulated, which led to the conclusion that g(t) is constrained to have a timescale of tau_tutor, which would mean that the main finding that your results "predict the temporal structure of the [LMAN signal] given a plasticity rule" – would be a trivial consequence of the plasticity rule….

However, as explained in a small sentence above Equation 1, what tau_tutor actually refers to is the timescale over which motor errors are integrated into the firing of tutor neurons. This definition of tau_tutor is specific to the Taylor series approximation taken in A.8 and hence, their matching results point to the robustness of the model. One way to summarize the finding is if you want to use an STDP rule in motor learning, you should convolve your motor error signal with a particular exponential smoothing function, with a particular timescale related to the plasticity rule, independent of the form of the motor error signal.

Another interesting observation is that their derivation for the timescale tau_tutor is not valid when tau_tutor is of a comparable magnitude to tau1 and tau2. Stated simply, they make an assumption in their derivation that tau_tutor >> tau1 and tau2. Therefore, they don't see clear matching in areas where this assumption is violated. A discussion of this in subsection “Match vs. unmatched learning” may also improve the readability of the paper.

The authors should comment on what they would conclude if one observed a different form for g(t) in the songbird brain. It's a subtle point – a g(t) with the \tau^* they predict would certainly shore up their theory, but one with a different form might mean either a) that their theory is wrong, or b) that their approximation (A.8) is wrong.

2) Both reviewers strongly agreed that more is needed to motivate what the model predicts. In the Abstract and Discussion, a claim is made that the results presented here make clear and testable predictions for experimentalists. The manuscript would be greatly improved if this were made more explicit and specific. Indeed, explicitly connecting the model to falsifiable experimental predictions would make the difference between this being a paper of truly broad interest.

Perhaps the authors envision revealing the precise form of the plasticity rule by looking at the structure of the tutor signal. Perhaps they envision inferring the form of the motor error signal. Perhaps they will test coarser effects that result from this kind of tutor signal, like the progressive clumping of student responses into burst-like events during learning. If these predictions were laid out more clearly, concerns about what exactly is being claimed through the calculation in Appendix 3 would be ameliorated.

A couple of other suggestions for how the manuscript might be revised are given here:

– The point \α = \β is the case that corresponds to the study by Mehaffey and Doupe, so a detailed discussion of the tutor \tau's that yield reasonable learning in that part of parameter space would be useful. How big is the range of \tau_{tutor}'s in that case? Does it span many orders of magnitude? How well does the model activity at these points align with real LMAN activity in the developing bird?

– What precise features of the firing of LMAN neurons are predicted for the models that yield efficient learning and match known plasticity features? What should an experimentalist measure in their spike trains? Putting some numbers to the predictions would be great.

3) One reviewer thought that Figure 3 is not very convincing that learning is good only along the diagonal. It does not seem to matter at short timescales (which the authors discuss briefly), where performance is equivalent regardless of whether timescales are matched or unmatched (e.g. 10 ms vs 80 ms). But even at timescales >80 ms, the performance does not seem strictly diagonal, and the asymptotic performance of the model is significantly worse than at shorter timescales. Therefore it seems like the two key findings – timescale matching and the ability to learn – do not coexist. The other reviewer noted that the error is on a log scale is very important here. There is a steep valley sloping towards the diagonal. The larger error reached at longer timescales was thus less worrisome. 3B shows that this amount of error doesn't stem from huge differences in the output time course. We would therefore like to ask the authors for another plot like B at yet longer timescales to see how and if things break down. Also, 250 iterations isn't that many in terms of number of renditions sung during development, and we might want to see what the error surface looks like after a few thousand. Finally, the paper does discuss the flattening of the valley at short timescales, but this could be expanded.

4. It seems misleading to call τ1 and τ2 the timescales of synaptic plasticity of the student. They are simply timescales of two exponential kernels that are linearly combined to create the final filtering kernel. If the authors want to relate τ1 and τ2 to the timescales of synaptic plasticity, they should include a discussion of how and why they consider them to be related. In general it would be helpful if the authors are more clear in defining their variables through the paper – especially since tau1 and tau2 are very different from tau_tutor.

[Editors' note: further revisions were requested prior to acceptance, as described below.]

Thank you for resubmitting your work entitled "Rules and mechanisms for efficient two-stage learning in neural circuits" for further consideration at *eLife*. Your revised article has been favorably evaluated by Andrew King (Senior editor), Reviewing editor Michael Frank, and one external reviewer.

The manuscript has been improved but there are some remaining issues that need to be addressed before acceptance, as outlined clearly by the reviewer below:

By and large we are happy with the changes the authors have made.

However, there are still a few important aspects of the paper that need to be significantly revised to improve clarity and address some issues of interpretation:

1) The (qualitative) definition of tau_tutor could still be much clearer. When we first read the initial submission, we mistakenly thought that tau_tutor represented the timescale over which the firing rate of the tutor neurons is modulated, rather than (as is actually the case) the timescale over which motor errors are integrated into the firing of tutor neurons. Based on our comments from last time, the authors have somewhat clarified this, however this could still be more clear (in fact, I got confused in the same way reading the revised paper). We think this could be easily clarified with a couple of minor changes

a) Just above Equation 4, revise to "Moreover, for effective learning, the timescale t_tutor – ***which quantifies the timescale on which error information is integrated into the tutor signal*** – appearing in Equation 3…" (or something similar).

b) Be a bit more specific when talking about "matching". For example, the first sentence in 2.3 simply reads "…when the tutor is matched to the student…". I think it would be much better (especially for less quantitative readers) to explicitly state what is being matched by expanding this to read something like "…when the ***timescale on which error information is integrated into the tutor signal (tau_tutor) is matched to the student plasticity rule***…".

c) More generally, throughout the paper replace the term "tutor timescale" – which many biologists will misinterpret as the timescale on which tutor activity varies – with "tutor error integration timescale" or something like this.

2) We appreciate the authors expanding the duration of simulations and range of timescales tested in the simulations shown in Figure 3 and Figure 5. However, the authors should be more explicit (and consistent) in explaining how these simulations are conducted. My understanding is that in Figure 3, tau_1 and tau_2 are fixed at [80 40]msec, and to obtain a particular t_tutor*, plasticity parameters α and β are adjusted for each simulation (subject to the constraint α-β=1). Provided this is correct, the fact that α and β were different for different simulations (i.e. for different squares in Figure 3, Figure 5) should be explicitly stated in the results text and/or legend to Figure 3. Similarly, as far as I can tell a similar approach (fix tau_1,2, vary α/β to get a particular tau_tutor*) was used in the analysis shown in Figure 5. However, this is not made clear in the text/caption (I find the caption to 5D especially confusing). Authors should explicitly state the procedures for 5D and explain any differences in the general methods used in 3C. Furthermore, the authors should present (maybe as a supplemental figure) the values of α or β used for all simulations (e.g. a heatmap of α values to go along with Figure 3 and Figure 5).

3) The following is a somewhat subtle point, so I'll leave it up to the Authors to deal with as they see fit.

As noted above, tau_tutor is the timescale over which motor errors are integrated into the firing of tutor neurons. Furthermore, the various models (linear rate, spiking, etc.) can work over a big range of tau_tutors, say from 10ms-1280msec, as shown in Figure 3 and Figure 5. In the revised Discussion, the authors correctly assert that LMAN/tutor firing "should reflect the integral of the motor error with the timescale predicted by the model". The authors also correctly point out that we don't really know what the motor error signal looks like.

However – a key physiological fact is that firing in the "tutor" brain area – LMAN – consists of both short bursts (~15 msec duration) as well as single spikes whose overall rate varies much more slowly. So, although we don't know how fast the error signal varies, even if the error signal were a pulse that only lasted, say, 1 msec, it would affect the firing of the LMAN neuron on timescale tau_tutor – e.g. over >100 msec if tau_tutor=100. So, if I'm getting this right, it seems implausible that a 15 msec burst in LMAN could possibly reflect motor error – even an infinitely brief motor error – if tau_tutor were longer than, say, 50 msec. The authors may want to discuss this issue, especially whether their model suggests that only slow changes in the rate of single-spikes, and not bursts, are carrying error information? I think that such an implication is both important for understanding the implications of the model and for guiding future work in the system.

Additionally, as indicated just below Equation 4, the model assumes that tau_tutor>>tau_1,2. In many simulations (definitely in Figure 3, also I think in Figure 5, see my question above), tau_1 and tau_2 have values of 40 and 80 msec, which are values derived from the Mehaffey and Doupe STDP paper. This would suggest that tau_tutor should be >>80 msec, and that cases with tau_tutor and tau_tutor* less than this value (which make up a significant region of the parameter space shown in Figure 3 and Figure 5) are biologically implausible. This seems to me to be something the authors should address.

---

## [Author Response]

*Essential revisions:*

*All involved found the work to be of high quality and of broad interest to researchers in sensorimotor and reinforcement learning, as well as being of particular interest to those working on neural recording in the songbird brain.*

*1) However, an invigorating consultation session among the reviewers was needed to get down to the bottom of whether some of the main modeling results were circular/trivial or not. It was determined that in fact they are not, and that rather some of them show that the model is robust to the precise form of the motor error. But the reason for this initial difference in opinion could be traced to a lack of clarity in the manuscript over the definition of tau_tutor. One reviewer originally understood it as the timescale over which the firing rate of the tutor neurons is modulated, which led to the conclusion that g(t) is constrained to have a timescale of tau_tutor, which would mean that the main finding that your results "predict the temporal structure of the [LMAN signal] given a plasticity rule" – would be a trivial consequence of the plasticity rule….*

*However, as explained in a small sentence above Equation 1, what tau_tutor actually refers to is the timescale over which motor errors are integrated into the firing of tutor neurons. This definition of tau_tutor is specific to the Taylor series approximation taken in A.8 and hence, their matching results point to the robustness of the model. One way to summarize the finding is if you want to use an STDP rule in motor learning, you should convolve your motor error signal with a particular exponential smoothing function, with a particular timescale related to the plasticity rule, independent of the form of the motor error signal.*

We are grateful for this suggestion. Indeed, upon re-reading we realized that the text could have been clearer on this point. Your suggested explanation is indeed clear and accurate and we have incorporated a version of this text below Equation 4.

*Another interesting observation is that their derivation for the timescale tau_tutor is not valid when tau_tutor is of a comparable magnitude to tau1 and tau2. Stated simply, they make an assumption in their derivation that tau_tutor >> tau1 and tau2. Therefore, they don't see clear matching in areas where this assumption is violated. A discussion of this in subsection “Match vs. unmatched learning” may also improve the readability of the paper.*

We have added a discussion of these assumptions in our derivation at the end of section “Learning in a rate-based model”. Since the plasticity rule smooths conductor inputs over timescales of order tau_1 and tau_2, tutor signals that vary on shorter timescales do not have a strong effect on learning, justifying the approximation. It is also possible to improve the approximation we are making by including more terms in the Taylor expansion that we use in section A.3. We added a note about this in the Discussion.

*The authors should comment on what they would conclude if one observed a different form for g(t) in the songbird brain. It's a subtle point – a g(t) with the \tau^* they predict would certainly shore up their theory, but one with a different form might mean either a) that their theory is wrong, or b) that their approximation (A.8) is wrong.*

This is a very good point, and we thank the reviewers for encouraging us to discuss this issue. We added a few sentences pointing out these different ways of interpreting a potential mismatch between theory and experiment in this case (Discussion section). We have also mentioned that we could improve the approximation by going to the next order in the Taylor series expansion that we used to derive the matched tutor.

*2) Both reviewers strongly agreed that more is needed to motivate what the model predicts. In the Abstract and Discussion, a claim is made that the results presented here make clear and testable predictions for experimentalists. The manuscript would be greatly improved if this were made more explicit and specific. Indeed, explicitly connecting the model to falsifiable experimental predictions would make the difference between this being a paper of truly broad interest.*

*Perhaps the authors envision revealing the precise form of the plasticity rule by looking at the structure of the tutor signal. Perhaps they envision inferring the form of the motor error signal. Perhaps they will test coarser effects that result from this kind of tutor signal, like the progressive clumping of student responses into burst-like events during learning. If these predictions were laid out more clearly, concerns about what exactly is being claimed through the calculation in Appendix 3 would be ameliorated.*

Our model provides a general method for calculating the tutor signal needed by a student circuit during learning, under the assumption that the student and tutor circuits have evolved together to maximize learning efficiency. This can be used to make predictions in several ways, as we now explain in the Discussion: (a) we could use recordings from tutor neurons together with measurements of motor error to infer the student plasticity rule; (b) we could conversely use the plasticity rule with the motor error to predict tutor spiking statistics; (c) we could electrically or optogenetically stimulate tutor neurons and test whether the student learns as predicted by our model.

*A couple of other suggestions for how the manuscript might be revised are given here:*

*– The point \α = \β is the case that corresponds to the study by Mehaffey and Doupe, so a detailed discussion of the tutor \tau's that yield reasonable learning in that part of parameter space would be useful. How big is the range of \tau_{tutor}'s in that case? Does it span many orders of magnitude? How well does the model activity at these points align with real LMAN activity in the developing bird?*

We added a comment about the range of tau_tutor’s that allow effective learning to the bottom of section “Matched vs. unmatched learning”. Because of the normalization we use, which sets α – β = 1, we cannot directly address the question of what happens when α = β. However, the Mehaffey & Doupe data can be explained just as well using a plasticity model that has α large, α>>1, which would set α and β approximately equal to each other (leading to a large tau_tutor^*). This implies that the Mehaffey & Doupe case corresponds to the bottom part of the plots in Figure 3.

A precise comparison of LMAN activity and our model would require identifying the specific muscle group to which the recorded LMAN neurons refer, which is feasible but hasn't been done yet. The relation between the muscle activations and song features (such as pitch) would also have to be known – again, this is possible in principle, but goes beyond the scope of our paper.

*– What precise features of the firing of LMAN neurons are predicted for the models that yield efficient learning and match known plasticity features? What should an experimentalist measure in their spike trains? Putting some numbers to the predictions would be great.*

Our model can predict how the average firing rate of LMAN neurons should vary during song production, provided we have a good measure of motor error and can estimate how specific LMAN neurons contribute to this error. Measurements of the latter kind are necessary for making specific quantitative comparisons, which thus lie beyond the scope of the present paper. Please also see the response to question (2.1), and the revised Discussion.

*3) One reviewer thought that Figure 3 is not very convincing that learning is good only along the diagonal. It does not seem to matter at short timescales (which the authors discuss briefly), where performance is equivalent regardless of whether timescales are matched or unmatched (e.g. 10 ms vs 80 ms). But even at timescales >80 ms, the performance does not seem strictly diagonal, and the asymptotic performance of the model is significantly worse than at shorter timescales. Therefore it seems like the two key findings – timescale matching and the ability to learn – do not coexist. The other reviewer noted that the error is on a log scale is very important here. There is a steep valley sloping towards the diagonal. The larger error reached at longer timescales was thus less worrisome. 3B shows that this amount of error doesn't stem from huge differences in the output time course. We would therefore like to ask the authors for another plot like B at yet longer timescales to see how and if things break down. Also, 250 iterations isn't that many in terms of number of renditions sung during development, and we might want to see what the error surface looks like after a few thousand. Finally, the paper does discuss the flattening of the valley at short timescales, but this could be expanded.*

We updated the heatmaps extending the range of timescales. Note that the timescales are chosen in geometric progression, so the four bins we added to each row and column of the heatmap extended the maximum tutor timescale by a factor of 2^4 = 16, to over 20 seconds. This is much longer than our typical simulation, which ran for about 1 second. We also ran the simulations for 1000 steps now instead of 250 in the previous version, and we used the same color scale throughout the paper to make it easier to make comparisons.

The new plots for the rate-based simulations show that both effective learning when the student matches the tutor, and impaired learning when there is a mismatch, persist even when the tutor timescales are very long. A more interesting effect occurs for the spiking simulations, where convergence breaks down for very long tutor memories. When the tutor timescale is significantly longer than the duration of the motor program, the tutor firing rates tend to stay close to the threshold theta. In the spiking case, the fluctuations due to Poisson spiking introduce noise in the tutor guiding signal, and this noise is more damaging to learning the less the tutor rates vary. This is one of the reasons for the failure in convergence in spiking simulations when the tutor timescale is very long. Another issue is related to the constraint that requires conductor-student synaptic weights to be positive. A consequence of this constraint is that negative fluctuations in the synaptic weights (which happen due to the stochasticity of the tutor signal) are sometimes clamped (when the weights reach zero), and so there is a net positive trend on the weights. This leads to a residual positive shift in the motor program that does not go away with learning.

Both effects can be reduced – increasing the gain that relates the motor error to the tutor signal reduces the effect of the fluctuations, and using a push-pull mechanism in which student neurons can either act to increase or decrease muscle activity effectively eliminates the residual shift in the motor program.

In the first version of the manuscript we had chosen to keep the tutor timescales in a range that didn’t significantly exceed the typical duration of the motor program, both because that seems to be the more relevant scenario (longer memory timescales do not have enough time to take full effect during the motor program), and to avoid the complications discussed above. We now included a discussion of these issues in section “Spiking neurons and birdsong”, and in the Supplementary Information, in section “Effect of spiking stochasticity on learning”.

*4. It seems misleading to call τ1 and τ2 the timescales of synaptic plasticity of the student. They are simply timescales of two exponential kernels that are linearly combined to create the final filtering kernel. If the authors want to relate τ1 and τ2 to the timescales of synaptic plasticity, they should include a discussion of how and why they consider them to be related. In general it would be helpful if the authors are more clear in defining their variables through the paper – especially since tau1 and tau2 are very different from tau_tutor.*

Indeed, we did not mean to imply that tau_1 and tau_2 are directly related to particular details of the plasticity mechanism, such as the duration it takes for plasticity to occur. In our model, tau_1 and tau_2 are simply two parameters that allow us to fit the observed dynamics of synaptic plasticity in some cases. We extended the description of our rule in section “Learning in a rate-based model” to make this clearer, and also to emphasize the relation between our rule and standard STDP rules. Rather than referring to these parameters as “plasticity timescales”, we now call them “student timescales”. We also added the equation defining the plasticity rule to subsection “Learning in a rate-based model” (Equation 1) to avoid confusion regarding the meaning of the different parameters used by the plasticity rule. We further adopted the notation tau_tutor to refer to the exponential kernel used to convolve the motor error in defining the tutor signal (Equation 2), and tau_tutor^* to refer to the timescale matched to a particular student.

[Editors' note: further revisions were requested prior to acceptance, as described below.]

*However, there are still a few important aspects of the paper that need to be significantly revised to improve clarity and address some issues of interpretation:*

*1) The (qualitative) definition of tau_tutor could still be much clearer. When we first read the initial submission, we mistakenly thought that tau_tutor represented the timescale over which the firing rate of the tutor neurons is modulated, rather than (as is actually the case) the timescale over which motor errors are integrated into the firing of tutor neurons. Based on our comments from last time, the authors have somewhat clarified this, however this could still be more clear (in fact, I got confused in the same way reading the revised paper). We think this could be easily clarified with a couple of minor changes*

*a) Just above Equation 4, revise to "Moreover, for effective learning, the timescale t_tutor – ***which quantifies the timescale on which error information is integrated into the tutor signal*** – appearing in Equation 3…" (or something similar).*

We added the suggested clarification above Equation 4.

*b) Be a bit more specific when talking about "matching". For example, the first sentence in 2.3 simply reads "…when the tutor is matched to the student…". I think it would be much better (especially for less quantitative readers) to explicitly state what is being matched by expanding this to read something like "…when the ***timescale on which error information is integrated into the tutor signal (tau_tutor) is matched to the student plasticity rule***…".*

We added the extended explanation of the matching at the top of subsection “Matched vs. unmatched learning”.

*c) More generally, throughout the paper replace the term "tutor timescale" – which many biologists will misinterpret as the timescale on which tutor activity varies – with "tutor error integration timescale" or something like this.*

We replaced the phrase “tutor timescale” by “tutor error integration timescale” throughout the manuscript.

*2) We appreciate the authors expanding the duration of simulations and range of timescales tested in the simulations shown in Figure 3 and Figure 5. However, the authors should be more explicit (and consistent) in explaining how these simulations are conducted. My understanding is that in Figure 3, tau_1 and tau_2 are fixed at [80 40]msec, and to obtain a particular t_tutor*, plasticity parameters α and β are adjusted for each simulation (subject to the constraint α-β=1). Provided this is correct, the fact that α and β were different for different simulations (i.e. for different squares in Figure 3, Figure 5) should be explicitly stated in the results text and/or legend to Figure 3. Similarly, as far as I can tell a similar approach (fix tau_1,2, vary α/β to get a particular tau_tutor*) was used in the analysis shown in Figure 5. However, this is not made clear in the text/caption (I find the caption to 5D especially confusing). Authors should explicitly state the procedures for 5D and explain any differences in the general methods used in 3C. Furthermore, the authors should present (maybe as a supplemental figure) the values of α or β used for all simulations (e.g. a heatmap of α values to go along with Figure 3 and Figure 5).*

Indeed, as the referee states, for the heatmaps the timescales tau_1 and tau_2 are kept fixed while α and β are varied, while keeping α-β = 1. This was already explained in the text in the Results section. In addition, we followed the referee’s suggestions and explicitly stated this again in the captions of Figure 3 and Figure 5. We further added a more detailed description of the values of α and β that we used in the supplementary information.

*3) The following is a somewhat subtle point, so I'll leave it up to the Authors to deal with as they see fit.*

*As noted above, tau_tutor is the timescale over which motor errors are integrated into the firing of tutor neurons. Furthermore, the various models (linear rate, spiking, etc.) can work over a big range of tau_tutors, say from 10ms-1280msec, as shown in Figure 3 and Figure 5. In the revised Discussion, the authors correctly assert that LMAN/tutor firing "should reflect the integral of the motor error with the timescale predicted by the model". The authors also correctly point out that we don't really know what the motor error signal looks like.*

*However – a key physiological fact is that firing in the "tutor" brain area – LMAN – consists of both short bursts (~15 msec duration) as well as single spikes whose overall rate varies much more slowly. So, although we don't know how fast the error signal varies, even if the error signal were a pulse that only lasted, say, 1 msec, it would affect the firing of the LMAN neuron on timescale tau_tutor – e.g. over >100 msec if tau_tutor=100. So, if I'm getting this right, it seems implausible that a 15 msec burst in LMAN could possibly reflect motor error – even an infinitely brief motor error – if tau_tutor were longer than, say, 50 msec. The authors may want to discuss this issue, especially whether their model suggests that only slow changes in the rate of single-spikes, and not bursts, are carrying error information? I think that such an implication is both important for understanding the implications of the model and for guiding future work in the system.*

If we have understood this comment correctly, the referee is asking whether variations of the LMAN signal over short timescales are informative about motor error when tau_tutor is long. While it is true that the timing of a single spike or short burst in any given trial is noisy and thus cannot hold precise information about rapid variations in the motor error, we assume that the effects of learning are averaged over many repetitions of the motor program. In this sense, learning ends up depending on average firing rates and not on the precise moments when spikes or bursts occur. Recordings suggest that the average firing rates are similar for spikes and bursts (Kao, Wright and Doupe, 2008). It is also true that short pulses in the error signal get convolved with an exponential decay with timescale tau_tutor in the tutor signal. However, a student circuit that is matched to the tutor circuit effectively performs the appropriate deconvolution to pick out the faster variations of the motor error even from slowly-varying tutor signals. This is assured by our matching condition from Equation 4, and is demonstrated by the results of our spiking simulations.

For the purposes of this manuscript, we chose to not explicitly take into account how the tutor signal is split between isolated spikes and bursts. This is because we focused on the effects from average tutor firing rates, which are independent of this distinction, and also because our model is applicable to systems other than the vocal production mechanism in songbirds, where the precise balance between isolated spiking and bursting may be different. It would nevertheless be very interesting to study the way in which tutor bursting affects learning in a model such as ours. We hope to look into these topics in future work.

Additionally, as indicated just below Equation 4, the model assumes that tau_tutor>>tau_1,2. In many simulations (definitely in Figure 3, also I think in Figure 5, see my question above), tau_1 and tau_2 have values of 40 and 80 msec, which are values derived from the Mehaffey and Doupe STDP paper. This would suggest that tau_tutor should be >>80 msec, and that cases with tau_tutor and tau_tutor* less than this value (which make up a significant region of the parameter space shown in Figure 3 and Figure 5) are biologically implausible. This seems to me to be something the authors should address.

Indeed the values of tau_1 and tau_2 were chosen here to qualitatively match the STDP curve from Mehaffey and Doupe when the ratio of α and β is approximately equal to 1. However, our model applies more generally, and hence we explore a wider range of parameters. Our analysis also applies to systems where two-stage learning happens in which the student plasticity can be modeled with different values for these timescales, or with a different ratio of α and β. This is why we consider a wide range of values for these parameters in our simulations.

Further, although our analytical derivation relies on the assumption tau_tutor>>tau_1,2, this is simply an approximation needed to make the calculations tractable. By running the simulations in parameter ranges in which this and other assumptions of our derivation are not valid, we can check the robustness of our matching condition beyond the range in which the strict derivation held true. Indeed, we find that many of the assumptions can apparently be relaxed without affecting the matching condition (such as replacing the rate-based dynamics by spiking), while others are more rigid (such as the condition tau_tutor>>tau_1,2). We have added a paragraph explaining these points towards the beginning of subsection “Matched vs. unmatched learning”.